# ONLINE RESTLESS BANDITS WITH UNOBSERVED STATES

## ABSTRACT

We study the online restless bandit problem, where each arm evolves according to a Markov chain independently, and the reward of pulling an arm depends on both the current state of the corresponding Markov chain and the action. The agent (decision maker) does not know the transition kernels and reward functions, and cannot observe the states of arms even after pulling. The goal is to sequentially choose which arms to pull so as to maximize the expected cumulative rewards collected. In this paper, we propose TSEETC, a learning algorithm based on Thompson Sampling with Episodic Explore-Then-Commit. The algorithm proceeds in episodes of increasing length and each episode is divided into exploration and exploitation phases. In the exploration phase in each episode, action-reward samples are collected in a round-robin way and then used to update the posterior as a mixture of Dirichlet distributions. At the beginning of the exploitation phase, TSEETC generates a sample from the posterior distribution as true parameters. It then follows the optimal policy for the sampled model for the rest of the episode. We establish the Bayesian regret bound $\tilde{\mathcal{O}}(\sqrt{T})$ for TSEETC, where $T$ is the time horizon. This is the first bound that is close to the lower bound of restless bandits, especially in an unobserved state setting. We show through simulations that TSEETC outperforms existing algorithms in regret.

## 1 INTRODUCTION

The restless multi-armed problem (RMAB) is a general setup to model many sequential decision making problems ranging from wireless communication (Tekin & Liu, 2011; Sheng et al., 2014), sensor/machine maintenance (Ahmad et al., 2009; Akbarzadeh & Mahajan, 2021) and healthcare (Mate et al., 2020; 2021). This problem considers one agent and $N$ arms. Each arm $i$ is modulated by a Markov chain $M^i$ with state transition function $P^i$ and reward function $R^i$. At each time, the agent decides which arm to pull. After the pulling, all arms undergo an action-dependent Markovian state transition. The goal is to decide which arm to pull to maximize the expected reward, i.e., $\mathbb{E}[\sum_{t=1}^{T} r_t]$, where $r_t$ is the reward at time $t$ and $T$ is the time horizon.

In this paper, we consider the online restless bandit problem with **unknown parameters (transition functions and reward functions)** and **unobserved states**. Many works concentrate on learning unknown parameters (Liu et al., 2010; 2011; Ortner et al., 2012; Wang et al., 2020; Xiong et al., 2022a;b) while ignoring the possibility that the states are also unknown. The unobserved states assumption is common in real-world applications, such as cache access (Paria & Sinha, 2021) and recommendation system (Peng et al., 2020). In the cache access problem, the user can only get the perceived delay but cannot know whether the requested content is stored in the cache before or after the access. Moreover, in the recommender system, we do not know the user's preference for the items. There are also some studies that consider the unobserved states. However, they often assume the parameters are known (Mate et al., 2020; Meshram et al., 2018; Akbarzadeh & Mahajan, 2021) and there is a lack of theoretical result (Peng et al., 2020; Hu et al., 2020). And the existing algorithms (Zhou et al., 2021; Jahromi et al., 2022) with theoretical guarantee do not match the lower regret bound of RMAB (Ortner et al., 2012).

One common way to handle the unknown parameters but with observed states is to use the optimism in the face of uncertainty (OFU) principle (Liu et al., 2010; Ortner et al., 2012; Wang et al., 2020). The regret bound in these works is too weak sometimes, because the baseline they consider, such

as pulling the fixed arms (Liu et al., 2010), is not optimal in RMAB problem. Ortner et al. (2012) derives the lower bound $\tilde{\mathcal{O}}(\sqrt{T})$ for RMAB problem. However, it is not clear whether there is an efficient computational method to search out the optimistic model in the confidence region (Lakshmanan et al., 2015). Another way to estimate the unknown parameters is Thompson Sampling (TS) method (Jung & Tewari, 2019; Jung et al., 2019; Jahromi et al., 2022; Hong et al., 2022). TS algorithm does not need to solve all instances that lie within the confident sets as OFU-based algorithms (Ouyang et al., 2017). What's more, empirical studies suggest that TS algorithms outperform OFU-based algorithms in bandit and Markov decision process (MDP) problems (Scott, 2010; Chapelle & Li, 2011; Osband & Van Roy, 2017).

Some studies assume that only the states of pulled arms are observable (Mate et al., 2020; Liu & Zhao, 2010; Wang et al., 2020; Jung & Tewari, 2019). They translate the partially observable Markov decision process (POMDP) problem into a fully observable MDP by regarding the state last observed and the time elapsed as a meta-state (Mate et al., 2020; Jung & Tewari, 2019), which is much simpler due to more observations about pulled arms. Mate et al. (2020), and Liu & Zhao (2010) derive the optimal index policy but they assume the known parameters. Restless-UCB in Wang et al. (2020) achieves the regret bound of $\tilde{\mathcal{O}}(T^{2/3})$, which does not match the lower bound $\tilde{\mathcal{O}}(\sqrt{T})$ regret, and also restricted to a specific Markov model. There are also some works that consider that the arm's state is not visible even after pulling (Meshram et al., 2018; Akbarzadeh & Mahajan, 2021; Peng et al., 2020; Hu et al., 2020; Zhou et al., 2021; Yemini et al., 2021) and the classic POMDP setting (Jahromi et al., 2022). However, there are still some challenges unresolved. Firstly, Meshram et al. (2018) and Akbarzadeh & Mahajan (2021) study the RMAB problem with unobserved states but with known parameters. However, the true value of the parameters are often unavailable in practice. Secondly, the works study RMAB from a learning perspective, e.g., Peng et al. (2020); Hu et al. (2020) but there are no regret analysis. Thirdly, existing policies with regret bound $\tilde{\mathcal{O}}(T^{2/3})$ (Zhou et al., 2021; Jahromi et al., 2022) often do not have a regret guarantee that scales as $\tilde{\mathcal{O}}(\sqrt{T})$, which is the lower bound in RMAB problem (Ortner et al., 2012). Yemini et al. (2021) considers the arms are modulated by two unobserved states and with linear reward. This linear structure is quite a bit of side information that the decision maker can take advantage of for decision making and problem-dependent $\log(T)$ is given.

To the best of our knowledge, there are no provably optimal policies that perform close to the offline optimum and match the lower bound in restless bandit, especially in unobserved states setting. The unobserved states bring much challenges to us. Firstly, we need to control estimation error about states, which itself is not directly observed. Secondly, the error depends on the model parameters in a complex way via Bayesian updating and the parameters are still unknown. Thirdly, since the state is not fully observable, the decision-maker cannot keep track of the number of visits to state-action pairs, a quantity that is crucial in the theoretical analysis. We design a learning algorithm TSEETC to estimate these unknown parameters, and benchmarked on a stronger oracle, we show that our algorithm achieves a tighter regret bound. In summary, we make the following contributions:

**Problem formulation.** We consider the online restless bandit problems with unobserved states and unknown parameters. Compared with Jahromi et al. (2022), our reward functions are unknown.

**Algorithmic design.** We propose TSEETC, a learning algorithm based on Thompson Sampling with Episodic Explore-Then-Commit. The whole learning horizon is divided into episodes of increasing length. Each episode is split into exploration and exploitation phases. In the exploration phase, to estimate the unknown parameters, we update the posterior distributions about unknown parameters as a mixture of Dirichlet distributions. For the unobserved states, we use the belief state to encode the historical information. In the exploitation phases, we sample the parameters from the posterior distribution and derive an optimal policy based on the sampled parameter. What's more, we design the determined episode length in an increasing manner to control the total episode number, which is crucial to bound the regret caused by exploration.

**Regret analysis.** We consider a stronger oracle which solves POMDP based on our belief state. And we define the pseudo-count to store the state-action pairs. Under a Bayesian framework, we show that the expected regret of TSEETC accumulated up to time $T$ is bounded by $\tilde{\mathcal{O}}(\sqrt{T})$, where $\tilde{\mathcal{O}}$ hides logarithmic factors. This bound improves the existing results (Zhou et al., 2021; Jahromi et al., 2022).

**Experiment results.** We conduct the proof-of-concept experiments, and compare our policy with existing baseline algorithms. Our results show that TSEETC outperforms existing algorithms and achieve a near-optimal regret bound.

## 2   RELATED WORK

We review the related works in two main domains: learning algorithm for unknown parameters, and methods to identify unknown states.

**Unknown parameters.** Since the system parameters are unknown in advance, it is essential to study RMAB problems from a learning perspective. Generally speaking, these works can be divided into two categories: OFU (Ortner et al., 2012; Wang et al., 2020; Xiong et al., 2022a; Zhou et al., 2021; Xiong et al., 2022b) or TS based (Jung et al., 2019; Jung & Tewari, 2019; Jahromi et al., 2022; Hong et al., 2022). The algorithms based on OFU often construct confidence sets for the system parameters at each time, find the optimistic estimator that is associated with the maximum reward, and then select an action based on the optimistic estimator. However, these methods may not perform close to the offline optimum because the baseline policy they consider, such as pulling only one arm, is often a heuristic policy and not optimal. In this case, the regret bound $\mathcal{O}(\log T)$ (Liu et al., 2010) is less meaningful. Apart from these works, posterior sampling (Jung & Tewari, 2019; Jung et al., 2019) were used to solve this problem. A TS algorithm generally samples a set of MDP parameters randomly from the posterior distribution, then actions are selected based on the sampled model. Jung & Tewari (2019) and Jung et al. (2019) provide theoretical guarantee $\tilde{\mathcal{O}}(\sqrt{T})$ in the Bayesian setting. TS algorithms are confirmed to outperform optimistic algorithms in bandit and MDP problems (Scott, 2010; Chapelle & Li, 2011; Osband & Van Roy, 2017).

**Unknown states.** There are some works that consider the states of the pulled arm are observed (Mate et al., 2020; Liu & Zhao, 2010; Wang et al., 2020; Jung & Tewari, 2019). Mate et al. (2020) and Liu & Zhao (2010) assumes the unobserved states but with known parameters. Wang et al. (2020) constructs an offline instance and give the regret bound $\tilde{\mathcal{O}}(T^{2/3})$. Jung & Tewari (2019) considers the episodic RMAB problems and the regret bound $\tilde{\mathcal{O}}(\sqrt{T})$ is guaranteed in the Bayesian setting. Some studies assume that the states are unobserved even after pulling. Akbarzadeh & Mahajan (2021) and Meshram et al. (2018) consider the RMAB problem with unknown states but known system parameters. And there is no regret guarantee. Peng et al. (2020) and Hu et al. (2020) consider the unknown parameters but there are also no any theoretical results. The most similar to our work is Zhou et al. (2021) and Jahromi et al. (2022). Zhou et al. (2021) considers that all arms are modulated by a common unobserved Markov Chain. They proposed the estimation method based on spectral method (Anandkumar et al., 2012) and learning algorithm based on upper confidence bound (UCB) strategy (Auer et al., 2002). They also give the regret bound $\tilde{\mathcal{O}}(T^{2/3})$ and there is a gap between the lower bound $\tilde{\mathcal{O}}(\sqrt{T})$ (Ortner et al., 2012). Jahromi et al. (2022) considers the POMDP setting and propose the pseudo counts to store the state-action pairs. Their learning algorithm is based on Ouyang et al. (2017) and the regret bound is also $\tilde{\mathcal{O}}(T^{2/3})$. And their algorithm is not programmable due to the pseudo counts is conditioned on the true counts which is uncountable.

## 3   PROBLEM SETTING

Consider a restless bandit problem with one agent and $N$ arms. Each arm $i \in [N] := \{1, 2, \ldots, N\}$ is associated with an independent discrete–time Markov chain $\mathcal{M}^i = (\mathcal{S}^i, P^i)$, where $\mathcal{S}^i$ is the state space and $P^i \in \mathbb{R}^{\mathcal{S}^i \times \mathcal{S}^i}$ the transition functions. Let $s_t^i$ denote the state of arm $i$ at time $t$ and $s_t = (s_t^1, s_t^2, \ldots, s_t^N)$ the state of all arms. Each arm $i$ is also associated with a reward functions $R^i \in \mathbb{R}^{\mathcal{S}^i \times \mathcal{R}}$, where $R^i(r \mid s)$ is the probability that the agent receives a reward $r \in \mathcal{R}$ when he pulls arm $i$ in state $s$. We assume the state spaces $\mathcal{S}^i$ and the reward set $\mathcal{R}$ are finite and known to the agent. The parameters $P^i$ and $R^i$, $i \in [N]$ are unknown, and the state $s_t$ is also unobserved to the agent. For the sake of notational simplicity, we assume that all arms have the same state spaces $\mathcal{S}$ with size $S$. Our result can be generalized in a straightforward way to allow different state spaces.

The whole game is divided into $T$ time steps. The initial state $s_1^i$ for each arm $i \in [N]$ is drawn from a distribution $h_i$ independently, which we assume to be known to the agent. At each time $t$, the agent chooses one arm $a_t \in [N]$ to pull and receives a reward $r_t \in \mathcal{R}$ with probability $R^{a_t}(r_t \mid s_t^{a_t})$. Note

that only the pulled arm has the reward feedback. His decision on which arm $a_t$ to pull is based on the observed history $\mathcal{H}_t = [a_1, r_1, a_2, r_2 \cdots, a_{t-1}, r_{t-1}]$. Note that the states of the arms are never observable, even after pulling. Each arm $i$ makes a state transition independently according to the associated $P^i$, whether it is pulled or not. This process continues until the end of the game. The goal of the agent is to maximize the total expected reward.

We use $\theta$ to denote the unknown $P^i$ and $R^i$ for $i \in [N]$ collectively. Since the true states are unobservable, the agent maintains a belief state $b_t^i = [b_t^i(s, \theta), s \in \mathcal{S}^i] \in \Delta_{\mathcal{S}^i}$ for each arm $i$, where

$$b_t^i(s, \theta) := \mathbb{P}\left(s_t^i = s \mid \mathcal{H}_t, \theta\right),$$

and $\Delta_{\mathcal{S}^i} := \left\{b \in \mathbb{R}_+^{\mathcal{S}^i} : \sum_{s \in \mathcal{S}^i} b(s) = 1\right\}$ is the probability simplex in $\mathbb{R}^{\mathcal{S}^i}$. Note that $b_t^i(s, \theta)$ depends on the unknown model parameter $\theta$, which itself has to be learned by the agent. We aggregate all arms as a whole Markov chain $\mathcal{M}$ and denote its transition matrix and reward function as $P$ and $R$, respectively. For a given $\theta$, the overall belief state $b_t = (b_t^1, b_t^2, \cdots, b_t^N)$ is a sufficient statistic for $\mathcal{H}_{t-1}$ (Smallwood & Sondik, 1973), so the agent can base his decision at time $t$ on $b_t$ only. Let $\Delta_b := \Delta_{\mathcal{S}^1} \times \cdots \times \Delta_{\mathcal{S}^N}$. A deterministic stationary policy $\pi : \Delta_b \to [N]$ maps a belief state to an action. The long-term average reward of a policy $\pi$ is defined as

$$J^\pi(h, \theta) := \limsup_{T \to \infty} \frac{1}{T} \mathbb{E}\left[\sum_{t=1}^T r_t \mid h, \theta\right]. \tag{1}$$

We use $J(h, \theta) = \sup_\pi J^\pi(h, \theta)$ to denote the optimal long-term average reward. We assume $J(h, \theta)$ is independent of the initial distribution $h$ as in Jahromi et al. (2022) and denoted it by $J(\theta)$. We make the following assumption.

**Assumption 1.** *The smallest element $\epsilon_1$ in the transition functions $P^i, i \in N$ is bigger than zero.*

**Assumption 2.** *The smallest element $\epsilon_2$ in the reward functions $R^i, i \in N$ is bigger than zero.*

Assumption 1 and Assumption 2 are strong in general, but they help us bound the error of belief estimation (De Castro et al., 2017). Assumption 1 also makes the MDP weakly communicating (Bertsekas et al., 2011). For weakly communicating MDP, it is known that there exists a bounded function $v(\cdot, \theta) : \Delta_b \to \mathbb{R}$ such that for all $b \in \Delta_b$ (Bertsekas et al., 2011),

$$J(\theta) + v(b, \theta) = \max_a \left\{r(b, a) + \sum_r P(r \mid b, a, \theta)v(b', \theta)\right\}, \tag{2}$$

where $v$ is the relative value function, $r(b, a) = \sum_s \sum_r b^a(s, \theta)R^a(r \mid s)r$ is the expected reward, $b'$ is the updated belief after obtaining the reward $r$, and $P(r \mid b, a, \theta)$ is the probability of observing $r$ in the next step, conditioned on the current belief $b$ and action $a$. The corresponding optimal policy is the maximizer of the right part in equation 2. Since the value function $v(, \theta)$ is finite, we can bound the span function $\mathrm{sp}(\theta) := \max_b v(b, \theta) - \min_b v(b, \theta)$ as Zhou et al. (2021). We show the details about this bound in Proposition 1 and denote the bound as $H$.

We consider the Bayesian regret. The parameters $\theta^*$ is randomly generated from a known prior distribution $Q$ at the beginning and then fixed but unknown to the agent. We measure the efficiency of a policy $\pi$ by its regret, defined as the expected gap between the cumulative reward of an offline oracle and that of $\pi$, where the oracle is the optimal policy with the full knowledge of $\theta^*$, but unknown states. The offline oracle is similar to Zhou et al. (2021), which is stronger than those considered in Azizzadenesheli et al. (2016) and Fiez et al. (2018). We focus on the Bayesian regret of policy $\pi$ (Ouyang et al., 2017; Jung & Tewari, 2019) as follows,

$$R_T := \mathbb{E}_{\theta^* \sim Q}\left[\sum_{t=1}^T (J(\theta^*) - r_t)\right]. \tag{3}$$

The above expectation is with respect to the prior distribution about $\theta^*$, the randomness in state transitions and the random reward.

## 4 THE TSEETC ALGORITHM

In section 4.1, we define the belief state and show how to update it with new observation. In section 4.2, we show how to update the posterior distributions under unknown states. In section 4.3, we show the details about our learning algorithm TSEETC.

### 4.1 Belief Encoder for Unobserved State

Here we focus on the belief update for arm $i$ with true parameters $\theta^*$. At time $t$, the belief for arm $i$ in state $s$ is $b_t^i(s, \theta^*)$. Then after the pulling of arm $i$, we obtain the observation $r_t$. The belief $b_t^i(s', \theta^*)$ can be update as follows:

$$b_{t+1}^i(s', \theta^*) = \frac{\sum_s b_t^i(s, \theta^*) R_*^i(r_t \mid s) P_*^i(s' \mid s)}{\sum_s b_t^i(s, \theta^*) R_*^i(r_t \mid s)}, \tag{4}$$

where the $P_*^i(s' \mid s)$ is the probability of transitioning from state $s$ at time $t$ to state $s'$ and $R_*^i(r_t \mid s)$ is the probability of obtain reward $r_t$ under state $s$.

If the arm $i$ is not pulled, we update its belief as follows:

$$b_{t+1}^i(s', \theta^*) = \sum_s b_t^i(s, \theta^*) P_*^i(s' \mid s). \tag{5}$$

Then at each time, we can aggregate the belief of all arms as $b_t$. Based on equation 2 , we can derive the optimal action $a_t$ for current belief $b_t$.

### 4.2 Mixture of Dirichlet Distribution

In this section, we estimate the unknown $P^i$ and $R^i$ based on Dirichlet distribution. The Dirichlet distribution is parameterized by a count vector, $\phi = (\phi_1, \dots, \phi_k)$, where $\phi_i \geq 0$, such that the density of probability distribution is defined as $f(p \mid \phi) \propto \prod_{i=1}^k p_i^{\phi_i - 1}$ (Ghavamzadeh et al., 2015).

Since the true states are unobserved, all state sequences should be considered, with some weight proportional to the likelihood of each state sequence (Ross et al., 2011). Denote the reward history collected from time $t_1$ till $t_2$ for arm $i$ as $r_{t_1:t_2}^i$ and similarly the states history is denoted as $s_{t_1:t_2}^i$. And the belief state history is denoted as $b_{t_1:t_2}^i$. Then with these history information, the posterior distribution $g_t(P^i)$ and $g_t(R^i)$ at time $t$ can be updated as in Lemma 1.

**Lemma 1.** *Under the unobserved state setting and assuming transition function $P^i$ with prior $g_0(P^i) = f(\frac{P^i - \epsilon_1 \mathbf{1}}{1 - \epsilon_1} \mid \phi^i)$ , reward function $R^i$ with prior $g_0(R^i) = f(\frac{R^i - \epsilon_2 \mathbf{1}}{1 - \epsilon_2} \mid \psi^i)$, with the information $r_{0:t}^i$ and $b_{0:t}^i$ , then the posterior distribution are as follows:*

$$g_t(P^i) \propto \sum_{\bar{s}_t^i \in \mathcal{S}_i^t} g_0(P^i) w(s_{0:t}^i) \prod_{s,s'} \left( \frac{P^i(s' \mid s) - \epsilon_1}{1 - \epsilon_1} \right)^{N_{s,s'}^i(\bar{s}_t^i) + \phi_{s,s'}^i - 1}, \tag{6}$$

$$g_t(R^i) \propto \sum_{\bar{s}_t^i \in \mathcal{S}_i^t} g_0(R^i) w(s_{0:t}^i) \prod_{s,r} \left( \frac{R^i(r \mid s) - \epsilon_2}{1 - \epsilon_2} \right)^{N_{s,r}^i(\bar{s}_t^i) + \psi_{s,r}^i - 1}. \tag{7}$$

*where $w(s_{0:t}^i)$ is the likelihood of state sequence $s_{0:t}^i$ and $\mathbf{1}$ is the vector with one in each position.*

The element $\mathbf{1}$ can be different lengths in correspondence with the dimension of $P$ and $R$. This procedure is summarized in Algorithm 1.

---

**Algorithm 1** Posterior Update for $R^i(s, \cdot)$ and $P^i(s, \cdot)$

---

1: Input: the history length $\tau_1$, the state space $\mathcal{S}_i$, the belief history $b_{0:\tau_1}^i$, the reward history $r_{0:\tau_1}^i$, the initial parameters $\phi_{s,s'}^i, \psi_{s,r}^i$, for $s, s' \in \mathcal{S}_i, r \in \mathcal{R}$,
2: generate $\mathcal{S}_i^{\tau_1}$ possible state sequences
3: calculate the weight $w(j) = \prod_{t=1}^{\tau_1} b_t^i(s, \theta), j \in \mathcal{S}_i^{\tau_1}$
4: **for** $j$ in $1, \dots, \mathcal{S}_i^{\tau_1}$ **do**
5:     count the occurence times of event $(s, s')$ and $(s, r)$ as $N_{s,s'}^i, N_{s,r}^i$ in sequence $j$
6:     update $\phi_{s,s'}^i \leftarrow \phi_{s,s'}^i + N_{s,s'}^i, \psi_{s,r}^i \leftarrow \psi_{s,r}^i + N_{s,r}^i$
7:     aggregate the $\phi_{s,s'}^i$ as $\phi(j), \psi_{s,r}^i$ as $\psi(j)$ for all $s, s' \in \mathcal{S}_i, r \in \mathcal{R}$
8: **end for**
9: update the mixture Dirichlet distribution
    $g_{\tau_1}(P^i) \propto \sum_{j=1}^{\mathcal{S}_i^{\tau_1}} w(j) f(\frac{P^i - \epsilon_1 \mathbf{1}}{1 - \epsilon_1} \mid \phi(j))$,
    $g_{\tau_1}(R^i) \propto \sum_{j=1}^{\mathcal{S}_i^{\tau_1}} w(j) f(\frac{R^i - \epsilon_2 \mathbf{1}}{1 - \epsilon_2} \mid \psi(j))$

---

With Algorithm 1, we can update the posterior distribution about the unknown parameters and sample from the posterior distribution as true parameters. The belief estimation error can be bounded by the distance between the sampled parameters and the true values (Proposition 2 ). The theoretical guarantee about estimation errors about unknown parameters is provided in Lemma 2.

### 4.3 OUR ALGORITHM

Our algorithm, TSEETC, operates in episodes with different lengths. Each episode is split into exploration phase and exploitation phase. Denote the episode number is $K_T$ and the first time in each episode is denoted as $t_k$. We use $T_k$ to denote the length of episode $k$ and it can be determined as: $T_k = T_1 + k - 1$, where $T_1 = \left\lceil \frac{\sqrt{T}+1}{2} \right\rceil$. The length of exploration phase in each episode is fixed as $\tau_1$ which satisfies $\tau_1 K_T = \mathcal{O}(\sqrt{T})$ and $\tau_1 \leq \frac{T_1 + K_T - 1}{2}$. With these notations, our whole algorithm is shown below.

---

**Algorithm 2** Thompson Sampling with Episodic Explore-Then-Commit

1: Input: prior $g_0(P)$,$g_0(R)$, initial belief $b_0$, exploration length $\tau_1$, the first episode length $T_1$
2: **for** episode $k = 1, 2, \ldots,$ **do**
3:     start the first time of episode $k$, $t_k := t$
4:     generate $R(t_k) \sim g_{t_{k-1}+\tau_1}(R)$ and $P(t_k) \sim g_{t_{k-1}+\tau_1}(P)$
5:     **for** $t = t_k, t_k + 1, ..., t_k + \tau_1$ **do**
6:         pull the arm $i$ for $\tau_1/N$ times in a round robin way
7:         receive the reward $r_t$
8:         update the belief $b_t^i$ using $R(t_k)$, $P(t_k)$ based on equation 4
9:         update the belief $b_t^j, j \in N \setminus \{i\}$ using $P(t_k)$ based on equation 5
10:     **end for**
11:     **for** $i = 1, 2, \ldots, N$ **do**
12:         input the obtained $r_{t_1:t_1+\tau_1}, ..., r_{t_k:t_k+\tau_1}$ , $b_{t_1:t_1+\tau_1}, ..., b_{t_k:t_k+\tau_1}$ to Algorithm 1 to update the posterior distribution $g_{t_k+\tau_1}(P), g_{t_k+\tau_1}(R)$
13:     **end for**
14:     generate $R(t_k + \tau_1) \sim g_{t_k+\tau_1}(P)$, $P(t_k + \tau_1) \sim g_{t_k+\tau_1}(R)$
15:     **for** $i$ in $0, 1, \ldots, N$ **do**
16:         re-update the belief $b_t^i$ from time 0 to $t_k + \tau_1$ based on $R(t_k + \tau_1)$ and $P(t_k + \tau_1)$
17:     **end for**
18:     compute $\pi_k^*(\cdot) = \text{Oracle}(\cdot, R(t_k + \tau_1), P(t_k + \tau_1))$
19:     **for** $t = t_k + \tau_1 + 1, \cdots, t_{k+1} - 1$ **do**
20:         apply action $a_t = \pi_k^*(b_t)$
21:         observe new reward $r_{t+1}$
22:         update the belief $b_t$ of all arms based equation 4, equation 5
23:     **end for**
24: **end for**

---

In episode $k$, for the exploration phase, we first sampled the $\theta_{t_k}$ from the distribution $g_{t_{k-1}+\tau_1}(P)$ and $g_{t_{k-1}+\tau_1}(R)$. We pull each arm for $\tau_1/N$ times in a round robin way. For the pulled arm, we update its belief based on equation 4 using $\theta_{t_k}$. For the arms that are not pulled, we update its belief based on equation 5 using $\theta_{t_k}$. The reward and belief history of each arm are input into Algorithm 1 to update the posterior distribution after the exploration phase. Then we sample the new $\theta_{t_k+\tau_1}$ from the posterior distribution, and re-calibrate the belief $b_t$ based on the most recent estimated $\theta_{t_k+\tau_1}$. Next we enter into the exploitation phase . Firstly we derive the optimal policy $\pi_k$ for the sampled parameter $\theta_{t_k+\tau_1}$. Then we use policy $\pi_k$ for the rest of the episode $k$.

We control the increasing of episode length in a deterministic manner. Specially, the length for episode $k$ is just one more than the last episode $k$. In such a deterministic increasing manner, the episode number $K_T$ is bounded by $\mathcal{O}(\sqrt{T})$ as in Lemma 10. Then the regret caused by the exploration phases can be bound by $\mathcal{O}(\sqrt{T})$, which is an crucial part in Theorem 1.

In TSEETC, for the unknown states, we propose the belief state to estimate the true states. What's more, under the unobserved state setting, we consider all possible state transitions and update the

posterior distribution of unknown parameters as mixture of each combined distribution, in which each occurence is summed with different weight.

**Remark 1.** *We use an Oracle to derive the optimal policy for the sampled parameters in Algorithm 2. The Oracle can be the Bellman equation for POMDP as we introduced in equation 2, or the approximation methods (Pineau et al., 2003; Silver & Veness, 2010), etc. The approximation error is discussed in Remark 3.*

## 5 PERFORMANCE ANALYSIS

In section 5.1, we show our theoretical results and some discussions. In section 5.2, we provide a proof sketch and the detailed proof is in Appendix B.

### 5.1 REGRET BOUND AND DISCUSSIONS

**Theorem 1.** *Suppose Assumptions 1,2 hold and the Oracle returns the optimal policy in each episode. The Bayesian regret of our algorithm satisfies*

$$R_T \leq 48 C_1 C_2 S \sqrt{NT \log(NT)} + (\tau_1 \Delta R + H + 4 C_1 C_2 SN) \sqrt{T} + C_1 C_2,$$

*where $C_1 = L_1 + L_2 N + N^2 + S^2, C_2 = r_{max} + H$ are constants independent with time horizon $T$, $L_1 = \frac{4(1-\epsilon_1)^2}{N \epsilon_1^2 \epsilon_2}, L_2 = \frac{4(1-\epsilon_1)^2}{\epsilon_1^3}$, $\epsilon_1$ and $\epsilon_2$ are the minimum elements of the functions $P^*$ and $R^*$, respectively. $\tau_1$ is the fixed exploration length in each episode, $\Delta R$ is the biggest gap of the reward obtained at each two different time, $H$ is the bounded span, $r_{max}$ is the maximum reward obtain each time, $N$ is the number of arms and $S$ is the state size for each arm.*

**Remark 2.** *The Theorem 1 shows that the regret of TSEETC is upper bound by $\tilde{\mathcal{O}}(\sqrt{T})$. This is the first bound that matches the lower bound in restless bandit problem (Ortner et al., 2012) in such unobserved state setting. Although TSEETC looks similar to explore-then-commit (Lattimore & Szepesvári, 2020), a key novelty of TSEETC lies in using the approach of posterior sampling to update the posterior distribution of unknown parameters as the mixture of each combined distribution. Our algorithm balances exploration and exploitation in a deterministic-episode manner and ensures the episode length grows at a linear rate, which guarantees that the total episode number is bounded by $\mathcal{O}(\sqrt{T})$. Therefore the total regret caused by exploration is well controlled by $\mathcal{O}(\sqrt{T})$ and this is better than the bound $\mathcal{O}(T^{2/3})$ in Zhou et al. (2021). What's more, in the exploitation phase, our regret bound $\tilde{\mathcal{O}}(\sqrt{T})$ is also better than $\tilde{\mathcal{O}}(T^{2/3})$ (Zhou et al., 2021). This shows our posterior sampling based method is superior to UCB based solution (Osband & Van Roy, 2017). In Jahromi et al. (2022), their pseudo count of state-action pair is always smaller than the true counts with some probability at any time. However, in our algorithm, the sampled parameter is more concentrated on true values with the posterior update. Therefore, our pseudo count (defined in equation 13) based on belief approximates the true counts more closely, which helps us obtain a tighter bound.*

### 5.2 PROOF SKETCH

In our algorithm, the total regret can be decomposed as follows:

$$R_T = \mathbb{E}_{\theta_*} \underbrace{\left[ \sum_{k=1}^{k_T} \sum_{t_k}^{t_k+\tau_1} J(\theta^*) - r_t \right]}_{\text{Regret (A)}} + \mathbb{E}_{\theta_*} \underbrace{\left[ \sum_{k=1}^{k_T} \sum_{t_k+\tau_1+1}^{t_{k+1}-1} J(\theta^*) - r_t \right]}_{\text{Regret (B)}}. \tag{8}$$

**Bounding Regret (A).** The Regret (A) is the regret caused in the exploration phase of each episode. This term can be simply bounded as follows:

$$\text{Regret (A)} \leq \mathbb{E}_{\theta_*} \left[ \sum_{k=1}^{k_T} \tau_1 \Delta R \right] \leq \tau_1 \Delta R k_T \tag{9}$$

where $\Delta R = r_{max} - r_{min}$ is the biggest gap of the reward received at each two different times. The regret in equation 9 is related with the episode number $k_T$, which can be bounded as $\mathcal{O}(\sqrt{T})$ in Lemma 10.

**Bounding Regret (B).** Next we bound Regret(B) in the exploitation phase. Define $\hat{b}_t$ is the belief updated with parameter $\theta_k$ and $b_t^*$ represents the belief with $\theta^*$. During episode $k$, based on equation 2 for the sampled parameter $\theta_k$ and that $a_t = \pi^*(\hat{b}_t)$, we can write:

$$J(\theta_k) + v(\hat{b}_t, \theta_k) = r(\hat{b}_t, a_t) + \sum_r P(r \mid \hat{b}_t, a_t, \theta_k) v(b', \theta_k). \tag{10}$$

With this equation, we proceed by decomposing the regret as:

$$\text{Regret(B)} = R_1 + R_2 + R_3 + R_4 \tag{11}$$

where each term is defined as follows:

$$R_1 = \mathbb{E}_{\theta^*} \sum_{k=1}^{k_T} \left[ (T_k - \tau_1 - 1)(J(\theta^*) - J(\theta_k)) \right],$$

$$R_2 = \mathbb{E}_{\theta^*} \sum_{k=1}^{k_T} \left[ \sum_{t_k+\tau_1+1}^{t_{k+1}-1} \left( v(\hat{b}_{t+1}, \theta_k) - v(\hat{b}_t, \theta_k) \right) \right],$$

$$R_3 = \mathbb{E}_{\theta^*} \sum_{k=1}^{k_T} \left[ \sum_{t_k+\tau_1+1}^{t_{k+1}-1} \left( \sum_r P\left[ r \mid \hat{b}_t, a_t, \theta_k \right] v(b', \theta_k) - v(\hat{b}_{t+1}, \theta_k) \right) \right],$$

$$R_4 = \mathbb{E}_{\theta^*} \sum_{k=1}^{k_T} \left[ \sum_{t_k+\tau_1+1}^{t_{k+1}-1} \left( r(\hat{b}_t, a_t) - r(b_t^*, a_t) \right) \right].$$

**Bounding $R_1$.** One key property of Posterior Sampling algorithms is that for given the history $\mathcal{H}_{t_k}$, the true parameter $\theta^*$ and sampled $\theta_k$ are identically distributed at the time $t_k$ as stated in Lemma 13. Due to the length $T_k$ determined and independent with $\theta_k$, then $R_1$ is zero thanks to this key property.

**Bounding $R_2$.** The regret $R_2$ is the telescopic sum of value function and can be bounded as $R_2 \leq H K_T$. It solely depends on the episode number and the upper bound $H$ of span function. As a result, $R_2$ reduce to a finite bound over the number of episodes $k_T$, which can be bounded in Lemma 10.

**Bounding $R_3$ and $R_4$.** The regret terms $R_3$ and $R_4$ is related with estimation error about $\theta$. Thus we should bound the parameters' error especially in our unobserved state setting. Recall the definition of $\phi, \psi$, we can define the posterior mean of $\hat{P}^i(s' \mid s)$ and $\hat{R}^i(r \mid s)$ for arm $i$ at time $t$ as follows:

$$\hat{P}^i(s' \mid s)(t) = \frac{\epsilon_1 + (1-\epsilon_1)\phi_{s,s'}^i(t)}{S\epsilon_1 + (1-\epsilon_1)\left\| \phi_{s,\cdot}^i(t) \right\|_1}, \quad \hat{R}^i(r \mid s)(t) = \frac{\epsilon_2 + (1-\epsilon_2)\psi_{s,r}^i(t)}{S\epsilon_2 + (1-\epsilon_2)\left\| \psi_{s,\cdot}^i(t) \right\|_1}. \tag{12}$$

We also define the pesudo count of the state-action pair $(s, a)$ before the episode $k$ as

$$N_{t_k}^i(s, a) = \left\| \psi_{s,\cdot}^i(t_k) \right\|_1 - \left\| \psi_{s,\cdot}^i(0) \right\|_1 \tag{13}$$

where $\psi_{s,\cdot}^i(t_k)$ represents the count of state-action $z = (s, a)$ pair before the episode $k$. Let $\mathcal{M}_k^i$ be the set of plausible MDPs in episode $k$ with reward function $R(r \mid z)$ and transition function $P(s' \mid z)$ satisfying,

$$\sum_{s' \in \mathcal{S}} \left| P(s' \mid z) - \hat{P}_k^i(s' \mid z) \right| \leq \beta_k(z), \quad \sum_{r \in \mathcal{R}} \left| R(r \mid z) - \hat{R}_k^i(r \mid z) \right| \leq \beta_k(z), \tag{14}$$

where $\beta_k^i(s, a) := \sqrt{\frac{14 S \log(2 N t_k T)}{\max\{1, N_{t_k}^i(s,a)\}}}$ is chosen conservatively (Auer et al., 2008) so that $\mathcal{M}_k^i$ contains both $P_*^i$ and $P_k^i$, $R_*^i$ and $R_k^i$ with high probability. $P_*^i$ and $R_*^i$ are the true parameters as we defined in section 4.1. Specially, for the unobserved state setting, the belief error under different parameters is upper bounded by the gap between the estimators as in Proposition 2. Then the core of the proofs lies in deriving a high-probability confidence set with our pesudo counts and show that the estimated error accumulated to $T$ for each arm is bounded by $\sqrt{T}$. Then with the error bound for each arm, we can derive the final error bound about the MDP aggregated by all arms as stated in Lemma 2.

**Lemma 2.** *(estimation errors) The total estimation error about transition functions accumulated by all exploitation phases satisfies the following bound*

$$\mathbb{E}_{\theta^*}\left[\sum_{k=1}^{K_T}\sum_{t=t_k+\tau_1+1}^{t_{k+1}-1}\|P^*-P_k\|_1\right] \le 48SN\sqrt{NT\log(NT)} + 4SN^2\sqrt{T} + N. \quad (15)$$

The Lemma 2 shows that the accumulated error is bounded by $\mathcal{O}(\sqrt{T})$, which is crucial to obtain the final bound as the observed-states setting (Ortner et al., 2012; Jung & Tewari, 2019). With $C_1 = L_1 + L_2N + N^2 + S^2$, We show the final bound about $R_3, R_4$ and the detailed proof in Appendix B.3,B.4.

**Lemma 3.** $R_3$ *satisfies the following bound*

$$R_3 \le 48C_1SH\sqrt{NT\log NT} + 4C_1SNH\sqrt{T} + C_1H.$$

**Lemma 4.** $R_4$ *satisfies the following bound*

$$R_4 \le 48C_1Sr_{max}\sqrt{NT\log(NT)} + 4C_1SNr_{max}\sqrt{T} + C_1r_{max}.$$

## 6 NUMERICAL EXPERIMENTS

In this section, we present proof-of-concept experiments and approximately implement TSEETC . We consider two arms and there are two hidden states for each arm. We pull just one arm each time. The learning horizon $T = 50000$, and each algorithm runs 100 iterations. The transition functions and reward functions for all arms are the same. We initialize the algorithm with uninformed Dirichlet prior on the unknown parameters. We compare our algorithm with simple heuristics $\epsilon$-greedy (Lattimore & Szepesvári, 2020) ($\epsilon = 0.01$), and Sliding-Window UCB (Garivier & Moulines, 2011) with specified window size, RUCB (Liu et al., 2010), Q-learning (Hu et al., 2020) and SEEU (Zhou et al., 2021). The results are shown in Figure 1. We can find that TSEETC has the minimum regret among these algorithms.

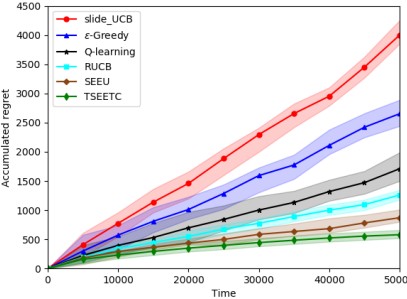

Figure 1: The cumulative regret

Figure 2: The log-log regret

In Figure 2, we plot the cumulative regret versus $T$ of the six algorithms in log-log scale. We observe that the slopes of all algorithms except for our TSEETC and SEEU are close to one, suggesting that they incur linear regrets. What is more, the slope of TSEETC is close to 0.5, which is better than SEEU. This is consistent with our theoretical result.

## 7 CONCLUSION

In this paper, we consider the restless bandit with unknown states and unknown dynamics. We propose the TSEETC algorithm to estimate these unknown parameters and derive the optimal policy. We also establish the Bayesian regret of our algorithm as $\tilde{\mathcal{O}}(\sqrt{T})$ which is the first bound that matches the lower bound especially in restless bandit problems with unobserved states . Numerical results validate that the TSEETC algorithm outperforms other learning algorithms in regret. A related open question is whether our method can be applied to the setting where the transition functions are action dependent. We leave it for future research.

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

## A  TABLE OF NOTATIONS

| Notation | Description |
|---|---|
| $T$ | The length of horizon |
| $K_T$ | The episode number of time $T$ |
| $T_k$ | The episode length of episode $k$ |
| $\tau_1$ | The fixed exploration length in each episode |
| $P^i$ | The transition functions for arm $i$ |
| $R^i$ | The reward function for arm $i$ |
| $P_k$ | The sampled transition function for aggregated MDP |
| $R_k$ | The sampled reward function for aggregated MDP |
| $r_t$ | The reward obtained at time $t$ |
| $b_t^i(s, \theta)$ | The belief state for being in state $s$ at time $t$ for arm $i$ with parameter $\theta$ |
| $\hat{b}_t$ | The belief of all arms at time $t$ with parameter $\theta_k$ |
| $b_t^*$ | The belief of all arms at time $t$ with parameter $\theta^*$ |
| $a_t$ | The action at time $t$ |
| $J(\theta_k)$ | The optimal long term average reward with parameter $\theta_k$ |
| $r_{max}$ | The maximum reward obtained each time |
| $r_{min}$ | The minimum reward obtained each time |
| $\Delta R$ | The biggest gap of the obtained reward |

## B  PROOF OF THEOREM 1

Recall that our goal is to minimize the regret :

$$R_T := \mathbb{E}_{\theta^*}\left[\sum_{t=1}^{T}(J(\theta^*) - r_t)\right]. \tag{16}$$

$r_t$ depends on the state $s_t$ and $a_t$. Thus $r_t$ can be written as $r(s_t, a_t)$. Due to $\mathbb{E}_{\theta^*}\left[r\left(s_t, a_t\right) \mid \mathcal{H}_{t-1}\right] = r(b_t^*, a_t)$ for any $t$, we have,

$$R_T := \mathbb{E}_{\theta^*}\left[\sum_{t=1}^{T}(J(\theta^*) - r(b_t^*, a_t))\right]. \tag{17}$$

In our algorithm, each episode is split into the exploration and exploitation phase then we can rewrite the regret as:

$$R_T = \mathbb{E}_{\theta^*}\left[\sum_{k=1}^{k_T}\sum_{t_k}^{t_k+\tau_1}(J(\theta^*) - r\left(b_t^*, a_t\right)) + \sum_{k=1}^{k_T}\sum_{t_k+\tau_1+1}^{t_{k+1}-1}(J(\theta^*) - r\left(b_t^*, a_t\right))\right], \tag{18}$$

where $\tau_1$ is the exploration length for each episode. $\tau_1$ is a constant. $t_k$ is the start time of episode $k$. Define the first part as Regret (A) which is caused by the exploration operations. The another part Regret (B) is as follows.

$$\text{Regret (A)} = \mathbb{E}_{\theta^*}\left[\sum_{k=1}^{k_T}\sum_{t_k}^{t_k+\tau_1}(J(\theta^*) - r\left(b_t^*, a_t\right))\right],$$

$$\text{Regret (B)} = \mathbb{E}_{\theta^*}\left[\sum_{k=1}^{k_T}\sum_{t_k+\tau_1+1}^{t_{k+1}-1}(J(\theta^*) - r\left(b_t^*, a_t\right))\right].$$

Recall that the reward set is $\mathcal{R}$ and we define the maximum reward gap in $\mathcal{R}$ as $\Delta R = r_{max} - r_{min}$. Then we get:

$$J(\theta^*) - r\left(b_t^*, a_t\right) \le \Delta R.$$

Then Regret (A) can be simply upper bounded as follows:

$$\text{Regret (A)} \le \mathbb{E}_{\theta^*}\left[\sum_{k=1}^{k_T}\tau_1\Delta R\right] \le \tau_1\Delta R k_T.$$

Regret (A) is related with the episode number $k_T$ obviously, which is bounded in Lemma 10. Next we should bound the term Regret (B).

During the episode $k$, based on equation 2, we get:

$$J\left(\theta_k\right) + v(\hat{b}_t, \theta_k) = r(\hat{b}_t, a_t) + \sum_r P(r \mid \hat{b}_t, a_t, \theta_k) v\left(b', \theta_k\right), \tag{19}$$

where $J\left(\theta_k\right)$ is the optimal long-term average reward when the system parameter is $\theta_k$, $\hat{b}_t$ is the belief at time $t$ updated with parameter $\theta_k$, $r(\hat{b}_t, a_t)$ is the expected reward we can get when the action $a_t$ is taken for the current belief $\hat{b}_t$, $b'$ is the updated belief based on equation 4 with parameter $\theta_k$ when the reward $r$ is received.

Using this equation, we proceed by decomposing the regret as:

$$\text{Regret(B)} = R_1 + R_2 + R_3 + R_4, \tag{20}$$

where

$$R_1 = \mathbb{E}_{\theta^*} \sum_{k=1}^{k_T} \left[(T_k - \tau_1 - 1)\left(J(\theta^*) - J(\theta_k)\right)\right],$$

$$R_2 = \mathbb{E}_{\theta^*} \sum_{k=1}^{k_T} \left[\sum_{t_k+\tau_1+1}^{t_{k+1}-1} \left(v(\hat{b}_{t+1}, \theta_k) - v(\hat{b}_t, \theta_k)\right)\right],$$

$$R_3 = \mathbb{E}_{\theta^*} \sum_{k=1}^{k_T} \left[\sum_{t_k+\tau_1+1}^{t_{k+1}-1} \left(\sum_r P(r \mid \hat{b}_t, a_t, \theta_k) v(b', \theta_k) - v(\hat{b}_{t+1}, \theta_k)\right)\right],$$

$$R_4 = \mathbb{E}_{\theta^*} \sum_{k=1}^{k_T} \left[\sum_{t_k+\tau_1+1}^{t_{k+1}-1} \left(r(\hat{b}_t, a_t) - r(b_t^*, a_t)\right)\right].$$

Next we bound the four parts one by one.

## B.1 Bound $R_1$

**Lemma 5.** $R_1$ *satisfies that* $R_1 = 0$.

*Proof.* Recall that:

$$R_1 = \mathbb{E}_{\theta^*} \sum_{k=1}^{k_T} \left[(T_k - \tau_1 - 1)\left(J(\theta^*) - J(\theta_k)\right)\right].$$

For each episode, $T_k$ is determined and is independent with $\theta_k$. Based on Lemma 13, we know that,

$$\mathbb{E}_{\theta^*}[J(\theta^*)] = \mathbb{E}_{\theta^*}[J(\theta_k)].$$

therefore, the part $R_1$ is 0.

$\square$

## B.2 Bound $R_2$

**Lemma 6.** $R_2$ *satisfies the following bound*

$$R_2 \le H K_T,$$

*where $K_T$ is the total number of episodes until time $T$.*

*Proof.* Recall that $R_2$ is the telescoping sum of value function at time $t + 1$ and $t$.

$$R_2 = \mathbb{E}_{\theta^*} \sum_{k=1}^{k_T} \left[\sum_{t=t_k+\tau_1+1}^{t_{k+1}-1} \left[v(\hat{b}_{t+1}, \theta_k) - v(\hat{b}_t, \theta_k)\right]\right]. \tag{21}$$

We consider the whole sum in episode $k$, then the $R_2$ can be rewrite as:

$$R_2 = \mathbb{E}_{\theta^*} \sum_{k=1}^{k_T} \left[ v(\hat{b}_{t_{k+1}}, \theta_k) - v(\hat{b}_{t_k+\tau_1+1}, \theta_k) \right].$$

Due to the span of $v(b, \theta)$ is bounded by $H$ as in proposition 1 , then we can obtain the final bound,

$$R_2 \leq HK_T.$$

$\square$

### B.3 BOUND $R_3$

In this section, we first rewrite the $R_3$ in section B.3.1. In section B.3.2, we show the details about how to bound $R_3$.

#### B.3.1 REWRITE $R_3$

**Lemma 7.** *(Rewrite $R_3$ ) The regret $R_3$ can be bounded as follows:*

$$R_3 \leq H\mathbb{E}_{\theta^*} \left[ \sum_{k=1}^{K_T} \sum_{t=t_k+\tau_1+1}^{t_{k+1}-1} ||P^* - P_k||_1 \right] + H\mathbb{E}_{\theta^*} \left[ \sum_{k=1}^{K_T} \sum_{t=t_k+\tau_1+1}^{t_{k+1}-1} ||b_t^* - \hat{b}_t||_1 \right]$$

$$+ S^2 H\mathbb{E}_{\theta^*} \left[ \sum_{k=1}^{k_T} \sum_{t=t_k+\tau_1+1}^{t_{k+1}-1} ||R^* - R_k||_1 \right],$$

*where $P_k$ is the sampled transition functions in episode $k$, $R_k$ is the sampled reward functions in episode $k$, $b_t^*$ is the belief at time $t$ updated with true $P^*$ and $R^*$, $\hat{b}_t$ is the belief at time $t$ updated with sampled $P_k, R_k$.*

*Proof.* The most part is similar to Jahromi et al. (2022), except that we should handle the unknown reward functions.

Recall that $R_3 = \mathbb{E}_{\theta^*} \sum_{k=1}^{k_T} \left[ \sum_{t=t_k+\tau_1+1}^{t_{k+1}-1} \left( \sum_r P(r \mid \hat{b}_t, a_t, \theta_k) v(b', \theta_k) - v(\hat{b}_{t+1}, \theta_k) \right) \right].$

Recall that $\mathcal{H}_t$ is the history of actions and observations prior to action $a_t$. Conditioned on $\mathcal{H}_t$, $\theta^*$ and $\theta_k$, the only random variable in $\hat{b}_{t+1}$ is $r_{t+1}$, then we can get,

$$\mathbb{E}_{\theta^*} \left[ v(\hat{b}_{t+1}, \theta_k) \mid \mathcal{H}_t, \theta_k \right] = \sum_{r \in R} v(b', \theta_k) P(r \mid b_t^*, a_t, \theta^*), \tag{22}$$

where $P(r \mid b_t^*, a_t, \theta^*)$ is the probability of getting reward $r$ given $b_t^*, a_t, \theta^*$. By the law of probability, $P(r \mid b_t^*, a_t, \theta^*)$ can be written as follows,

$$P(r \mid b_t^*, a_t, \theta^*) = \sum_{s'} R^*(r \mid s') P(s_{t+1} = s' \mid \mathcal{H}_t, \theta^*)$$

$$= \sum_{s'} R^*(r \mid s') \sum_s P^*(s_{t+1} = s' \mid s_t = s, \mathcal{H}_t, a_t, \theta^*) P(s_t = s \mid \mathcal{H}_t, \theta^*)$$

$$= \sum_s \sum_{s'} b_t^*(s) P^*(s' \mid s) R^*(r \mid s'),$$

$$\tag{23}$$

where $P^*$ is the transition functions for the MDP aggregated by all arms, $R^*$ is the reward function for the aggregated MDP. Therefore, we can rewrite the $R_3$ as follows,

$$R_3 = \mathbb{E}_{\theta^*} \left[ \sum_{k=1}^{k_T} \sum_{t=t_k+\tau_1+1}^{t_{k+1}-1} \left( \sum_{r \in R} (P(r \mid \hat{b}_t, a_t, \theta_k) - P(r \mid b_t^*, a_t, \theta^*) v(b', \theta_k) \right) \right].$$

Based on equation 23, we get

$$
R_3 = \mathbb{E}_{\theta^*} \left[ \sum_{k=1}^{k_T} \sum_{t=t_k+\tau_1+1}^{t_{k+1}-1} \left( \sum_r \sum_{s'} v(b', \theta_k) R_k(r \mid s') \sum_s \hat{b}_t(s) P_k(s' \mid s) \right) \right]
$$
$$
- \mathbb{E}_{\theta^*} \left[ \sum_{k=1}^{k_T} \sum_{t=t_k+\tau_1+1}^{t_{k+1}-1} \left( \sum_r \sum_{s'} v(b', \theta_k) R^*(r \mid s') \sum_s b_t^*(s) P^*(s' \mid s) \right) \right]
$$

$$
= \mathbb{E}_{\theta^*} \left[ \sum_{k=1}^{k_T} \sum_{t=t_k+\tau_1+1}^{t_{k+1}-1} \left( \sum_r \sum_{s'} v(b', \theta_k) R_k(r \mid s') \sum_s \hat{b}_t(s) P_k(s' \mid s) \right) \right]
$$
$$
- \mathbb{E}_{\theta^*} \left[ \sum_{k=1}^{k_T} \sum_{t=t_k+\tau_1+1}^{t_{k+1}-1} \left( \sum_r \sum_{s'} v(b', \theta_k) R_k(r \mid s') \sum_s b_t^*(s) P^*(s' \mid s) \right) \right] \tag{24}
$$
$$
+ \mathbb{E}_{\theta^*} \left[ \sum_{k=1}^{k_T} \sum_{t=t_k+\tau_1+1}^{t_{k+1}-1} \left( \sum_r \sum_{s'} v(b', \theta_k) R_k(r \mid s') \sum_s b_t^*(s) P^*(s' \mid s) \right) \right]
$$
$$
- \mathbb{E}_{\theta^*} \left[ \sum_{k=1}^{k_T} \sum_{t=t_k+\tau_1+1}^{t_{k+1}-1} \left( \sum_r \sum_{s'} v(b', \theta_k) R^*(r \mid s') \sum_s b_t^*(s) P^*(s' \mid s) \right) \right].
$$

where $R_k$ is the sampled reward function for aggregated MDP, $P_k$ is the sampled transition function for aggregated MDP.

Define
$$
R_3' = \mathbb{E}_{\theta^*} \left[ \sum_{k=1}^{k_T} \sum_{t=t_k+\tau_1+1}^{t_{k+1}-1} \left( \sum_r \sum_{s'} v(b', \theta_k) R_k(r \mid s') \left[ \sum_s \hat{b}_t(s) P_k(s' \mid s) - \sum_s b_t^*(s) P^*(s' \mid s) \right] \right) \right],
$$
$$
R_3'' = \mathbb{E}_{\theta^*} \left[ \sum_{k=1}^{k_T} \sum_{t=t_k+\tau_1+1}^{t_{k+1}-1} \left( \sum_r \sum_{s'} v(b', \theta_k) [R_k(r \mid s') - R^*(r \mid s')] \sum_s b_t^*(s) P^*(s' \mid s) \right) \right].
$$

**Bounding $R_3'$.** The part $R_3'$ can be bounded as Jahromi et al. (2022).

$$
R_3' = \mathbb{E}_{\theta^*} \left[ \sum_{k=1}^{k_T} \sum_{t=t_k+\tau_1+1}^{t_{k+1}-1} \left( \sum_r \sum_{s'} v(b', \theta_k) R_k(r \mid s') \left[ \sum_s \hat{b}_t(s) P_k(s' \mid s) - \sum_s b_t^*(s) P^*(s' \mid s) \right] \right) \right]
$$
$$
= R_3'(0) + R_3'(1)
$$

where

$$
R_3'(0) = \mathbb{E}_{\theta^*} \left[ \sum_{k=1}^{k_T} \sum_{t=t_k+\tau_1+1}^{t_{k+1}-1} \left( \sum_r \sum_{s'} v(b', \theta_k) R_k(r \mid s') \sum_s \hat{b}_t(s) P_k(s' \mid s) \right) \right]
$$
$$
- \mathbb{E}_{\theta^*} \left[ \sum_{k=1}^{k_T} \sum_{t=t_k+\tau_1+1}^{t_{k+1}-1} \left( \sum_r \sum_{s'} v(b', \theta_k) R_k(r \mid s') \sum_s b_t^*(s) P_k(s' \mid s) \right) \right]
$$

$$
R_3'(1) = \mathbb{E}_{\theta^*} \left[ \sum_{k=1}^{k_T} \sum_{t=t_k+\tau_1+1}^{t_{k+1}-1} \left( \sum_r \sum_{s'} v(b', \theta_k) R_k(r \mid s') \sum_s b_t^*(s) P_k(s' \mid s) \right) \right]
$$
$$
- \mathbb{E}_{\theta^*} \left[ \sum_{k=1}^{k_T} \sum_{t=t_k+\tau_1+1}^{t_{k+1}-1} \left( \sum_r \sum_{s'} v(b', \theta_k) R_k(r \mid s') \sum_s b_t^*(s) P^*(s' \mid s) \right) \right]
$$

For $R'_3(0)$, because $\sum_r R_k(r \mid s') = 1, \sum_{s'} P_k(s' \mid s) = 1, v(b', \theta_k) \leq H$, we have

$$R'_3(0) = \mathbb{E}_{\theta^*} \left[ \sum_{k=1}^{k_T} \sum_{t=t_k+\tau_1+1}^{t_{k+1}-1} \left( \sum_r \sum_{s'} v(b', \theta_k) R_k(r \mid s') \sum_s \hat{b}_t(s) P_k(s' \mid s) \right) \right]$$

$$- \mathbb{E}_{\theta^*} \left[ \sum_{k=1}^{k_T} \sum_{t=t_k+\tau_1+1}^{t_{k+1}-1} \left( \sum_r \sum_{s'} v(b', \theta_k) R_k(r \mid s') \sum_s b_t^*(s) P_k(s' \mid s) \right) \right]$$

$$= \mathbb{E}_{\theta^*} \left[ \sum_{k=1}^{k_T} \sum_{t=t_k+\tau_1+1}^{t_{k+1}-1} \left( \sum_r \sum_{s'} v(b', \theta_k) R_k(r \mid s') \sum_s (\hat{b}_t(s) - b_t^*(s) P_k(s' \mid s)) \right) \right]$$

$$\leq \mathbb{E}_{\theta^*} \left[ \sum_{k=1}^{k_T} \sum_{t=t_k+\tau_1+1}^{t_{k+1}-1} \left( \sum_r \sum_{s'} v(b', \theta_k) R_k(r \mid s') \sum_s |\hat{b}_t(s) - b_t^*(s)| P_k(s' \mid s) \right) \right]$$

$$\leq H \mathbb{E}_{\theta^*} \left[ \sum_{k=1}^{k_T} \sum_{t=t_k+\tau_1+1}^{t_{k+1}-1} \left( \sum_s |\hat{b}_t(s) - b_t^*(s)| \right) \right]$$

$$= H \mathbb{E}_{\theta^*} \left[ \sum_{k=1}^{k_T} \sum_{t=t_k+\tau_1+1}^{t_{k+1}-1} \left( \left\| \hat{b}_t(s) - b_t^*(s) \right\|_1 \right) \right],$$

where the first inequality is due to $\hat{b}_t(s) - b_t^*(s) \leq |\hat{b}_t(s) - b_t^*(s)|$ and the second inequality is because $\sum_r R_k(r \mid s') = 1, \sum_{s'} P_k(s' \mid s) = 1, v(b', \theta_k) \leq H$.

For the first term in $R'_3(1)$, note that conditioned on $\mathcal{H}_t, \theta^*$, the distribution of $s_t$ is $b_t^*$. Furthermore, $a_t$ is measurable with respect to the sigma algebra generated by $\mathcal{H}_t, \theta_k$ since $a_t = \pi^*(\hat{b}_t, \theta_k)$. Thus, we have

$$\mathbb{E}_{\theta^*} \left[ v(b', \theta_k) \sum_s P^*(s' \mid s) b^*(s) \mid H_t, \theta_k \right] = v(b', \theta_k) \mathbb{E}_{\theta^*} [P^*(s' \mid s) \mid H_t, \theta_k]. \tag{25}$$

$$\mathbb{E}_{\theta^*} \left[ v(b', \theta_k) \sum_s P_k(s' \mid s) b^*(s) \mid H_t, \theta_k \right] = v(b', \theta_k) \mathbb{E}_{\theta^*} [P_k(s' \mid s) \mid H_t, \theta_k]. \tag{26}$$

Substitute equation 25, equation 26 into $R'_3(1)$, we have

$$R'_3(1) = \mathbb{E}_{\theta^*} \left[ \sum_{k=1}^{k_T} \sum_{t=t_k+\tau_1+1}^{t_{k+1}-1} \left( \sum_r \sum_{s'} v(b', \theta_k) R_k(r \mid s') (P_k(s' \mid s) - P^*(s' \mid s)) \right) \right]$$

$$\leq \mathbb{E}_{\theta^*} \left[ \sum_{k=1}^{k_T} \sum_{t=t_k+\tau_1+1}^{t_{k+1}-1} \left( \sum_r \sum_{s'} v(b', \theta_k) R_k(r \mid s') |P_k(s' \mid s) - P^*(s' \mid s)| \right) \right]$$

$$\leq H \mathbb{E}_{\theta^*} \left[ \sum_{k=1}^{k_T} \sum_{t=t_k+\tau_1+1}^{t_{k+1}-1} \left( \sum_{s'} |P_k(s' \mid s) - P^*(s' \mid s)| \right) \right]$$

$$\leq H \mathbb{E}_{\theta^*} \left[ \sum_{k=1}^{k_T} \sum_{t=t_k+\tau_1+1}^{t_{k+1}-1} (\|P_k - P^*\|_1) \right],$$

where the first inequality is because $P_k(s' \mid s) - P^*(s' \mid s) \leq |P_k(s' \mid s) - P^*(s' \mid s)|$, the second inequality is due to $v(b', \theta_k) \leq H$ and $\sum_r R_k(r \mid s') = 1$.

Therefore we obtain the final results,

$$R'_3 \leq H \mathbb{E} \left[ \sum_{k=1}^{K_T} \sum_{t=t_k+\tau_1+1}^{t_{k+1}-1} \|P^* - P_k\|_1 \right] + H \mathbb{E} \left[ \sum_{k=1}^{K_T} \sum_{t=t_k+\tau_1+1}^{t_{k+1}-1} \|b_t^* - \hat{b}_t\|_1 \right].$$

**Bounding $R_3''$.** For part $R_3''$, note that for any fixed $s'$, $\sum_s b_t^*(s)P^*(s' \mid s) \leq S$, therefore we can bound $R_3''$ as follows,

$$
\begin{aligned}
R_3'' &= \mathbb{E}_{\theta^*}\left[\sum_{k=1}^{k_T}\sum_{t=t_k+\tau_1+1}^{t_{k+1}-1}\left(\sum_r\sum_{s'}v(b',\theta_k)\left[R_k(r\mid s')-R^*(r\mid s')\right]\sum_s b_t^*(s)P^*(s'\mid s)\right)\right]\\
&\leq SH\mathbb{E}_{\theta^*}\left[\sum_{k=1}^{k_T}\sum_{t=t_k+\tau_1+1}^{t_{k+1}-1}\left(\sum_{s'}\sum_r\left[R_k(r\mid s')-R^*(r\mid s')\right]\right)\right]\\
&\leq SH\mathbb{E}_{\theta^*}\left[\sum_{k=1}^{k_T}\sum_{t=t_k+\tau_1+1}^{t_{k+1}-1}S\left\|R_k-R^*\right\|_1\right]\\
&\leq S^2H\mathbb{E}_{\theta^*}\left[\sum_{k=1}^{k_T}\sum_{t=t_k+\tau_1+1}^{t_{k+1}-1}\left\|R_k-R^*\right\|_1\right],
\end{aligned}
$$

(27)

where the first inequality is due to $v(b',\theta_k) \leq H$ and $\sum_s b_t^*(s)P^*(s' \mid s) \leq S$, the second inequality is due to for any fixed $s'$, $\sum_r\left[R_k(r\mid s')-R^*(r\mid s')\right] \leq \|R_k-R^*\|_1$.

### B.3.2 Bound $R_3$

**Lemma 8.** $R_3$ satisfies the following bound

$$
\begin{aligned}
R_3 &\leq 48(L_1+L_2N+N+S^2)SH\sqrt{NT\log(NT)}+(L_1+L_2N+N+S^2)H\\
&\quad + 4(L_1+L_2N+N^2+S^2)SNH(T_1+K_T-\tau_1-1).
\end{aligned}
$$

*Proof.* Recall that the $R_3$ is as follows:

$$
R_3 = \mathbb{E}_{\theta^*}\sum_{k=1}^{k_T}\left[\sum_{t=t_k+\tau_1+1}^{t_{k+1}-1}\left(\sum_r P[r\mid\hat{b}_t,a_t,\theta_k]v(b',\theta_k)-v(\hat{b}_{t+1},\theta_k)\right)\right].
$$

This regret terms are dealing with the model estimation errors. That is to say, they depend on the on-policy error between the sampled transition functions and the true transition functions, the sampled reward functions and the true reward functions. Thus we should bound the parameters' error especially in our unobserved state setting. Based on the parameters in our Dirichlet distribution, we can define the empirical estimation of reward function and transition functions for arm $i$ as follows:

$$
\hat{P}^i(s'\mid s)(t) = \frac{\epsilon_1+(1-\epsilon_1)\phi_{s,s'}^i(t)}{S\epsilon_1+(1-\epsilon_1)\left\|\phi_{s,\cdot}^i(t)\right\|_1}, \quad \hat{R}^i(r\mid s)(t) = \frac{\epsilon_2+(1-\epsilon_2)\psi_{s,r}^i(t)}{S\epsilon_2+(1-\epsilon_2)\left\|\psi_{s,\cdot}^i(t)\right\|_1}. \quad (28)
$$

where $\phi_{s,s'}^i(t)$ is the parameters in the posterior distribution of $P^i$ at time $t$, $\psi_{s,r}^i(t)$ is the parameters in the posterior distribution of $R^i$ at time $t$. We also define the pseudo count $N_{t_k}^i(s,a)$ of the state-action pair $(s,a)$ before the episode $k$ for arm $i$ as

$$
N_{t_k}^i(s,a) = \left\|\psi_{s,\cdot}^i(t_k)\right\|_1 - \left\|\psi_{s,\cdot}^i(0)\right\|_1.
$$

For notational simplicity, we use $z = (s,a) \in \mathcal{S}\times\mathcal{A}$ and $z_t = (s_t,a_t)$ to denote the corresponding state-action pair. Then based on Lemma 7 we can decompose the $R_3$ as follows,

$$
\begin{aligned}
R_3 &= \mathbb{E}_{\theta^*}\left[\sum_{k=1}^{k_T}\sum_{t=t_k+\tau_1+1}^{t_{k+1}-1}\left(\sum_r P[r\mid\hat{b}_t,a_t,\theta_k]v(b',\theta_k)-v(\hat{b}_{t+1},\theta_k)\right)\right]\\
&= \mathbb{E}_{\theta^*}\left[\sum_{k=1}^{K_T}\sum_{t=t_k+\tau_1+1}^{t_{k+1}-1}\left[\sum_r\left(P(r\mid\hat{b}_t,a_t,\theta_k)-P(r\mid b_t^*,a_t,\theta^*)\right)v(b',\theta_k)\right]\right]\\
&\leq R_3^0+R_3^1+R_3^2
\end{aligned}
$$

where

$$R_3^0 = H\mathbb{E}_{\theta^*}\left[\sum_{k=1}^{K_T}\sum_{t=t_k+\tau_1+1}^{t_{k+1}-1}\|P^* - P_k\|_1\right],$$

$$R_3^1 = H\mathbb{E}_{\theta^*}\left[\sum_{k=1}^{K_T}\sum_{t=t_k+\tau_1+1}^{t_{k+1}-1}\|b_t^* - \hat{b}_t\|_1\right],$$

$$R_3^2 = S^2 H\mathbb{E}_{\theta^*}\left[\sum_{k=1}^{K_T}\sum_{t=t_k+\tau_1+1}^{t_{k+1}-1}\|R^* - R_k\|_1\right].$$

Note that the following results are all focused on one arm. Define $P_*^i$ is the true transition function for arm $i$, $P_k^i$ is the sampled transition function for arm $i$. We can extend the results on a arm to the aggregated large MDP based on Lemma 11.

**Bounding $R_3^0$.** Since $0 \le v(b', \theta_k) \le H$ from our assumption , each term in the inner summation is bounded by

$$\sum_{s'\in\mathcal{S}}\left|\left(P_*^i(s'\mid z_t) - P_k^i(s'\mid z_t)\right)\right|v(s', \theta_k)$$

$$\le H\sum_{s'\in\mathcal{S}}\left|P_*^i(s'\mid z_t) - P_k^i(s'\mid z_t)\right|$$

$$\le H\sum_{s'\in\mathcal{S}}\left|P_*^i(s'\mid z_t) - \hat{P}_k^i(s'\mid z_t)\right| + H\sum_{s'\in\mathcal{S}}\left|P_k^i(s'\mid z_t) - \hat{P}_k^i(s'\mid z_t)\right|.$$

where $P_*^i(s'\mid z_t)$ is the true transition function, $P_k^i(s'\mid z_t)$ is the sampled reward function and $\hat{P}_k^i(s'\mid z_t)$ is the posterior mean. The second inequality above in due to triangle inequality. Let $\mathcal{M}_k^i$ be the set of plausible MDPs in episode $k$ with reward function $R(r\mid z)$ and transition function $P(s'\mid z)$ satisfying,

$$\sum_{s'\in\mathcal{S}}\left|P(s'\mid z) - \hat{P}_k^i(s'\mid z)\right| \le \beta_k^i(z), \quad \sum_{r\in\mathcal{R}}\left|R(r\mid z) - \hat{R}_k^i(r\mid z)\right| \le \beta_k^i(z),$$

where $\beta_k^i(s, a) := \sqrt{\frac{14S\log(2Nt_kT)}{\max\{1, N_{t_k}^i(s,a)\}}}$ is chosen conservatively (Auer et al., 2008) so that $\mathcal{M}_k^i$ contains both $P_*^i$ and $P_k^i$, $R_*^i$ and $R_k^i$ with high probability. $P_*^i$ and $R_*^i$ are the true parameters as we defined in section 4.1. Note that $\beta_k^i(z)$ is the confidence set with $\delta = 1/t_k$. Recall the definition of $\psi$, we can define the pseudo count of state-action pair $(s, a)$ as $N_{t_k}^i(s, a) = \left\|\psi_{s,\cdot}^i(t_k)\right\|_1 - \left\|\psi_{s,\cdot}^i(0)\right\|_1$. Then we can obtain,

$$\sum_{s'\in\mathcal{S}}\left|P_*^i(s'\mid z_t) - \hat{P}_k^i(s'\mid z_t)\right| + \sum_{s'\in\mathcal{S}}\left|P_k^i(s'\mid z_t) - \hat{P}_k^i(s'\mid z_t)\right|$$

$$\le 2\beta_k^i(z_t) + 2\left(\mathbb{I}_{\{P_*^i\notin B_k\}} + \mathbb{I}_{\{P_k^i\notin B_k\}}\right). \tag{29}$$

We assume the length of the last episode is the biggest. Note that even the assumption does not hold, we can enlarge the sum items as $T_{K_T-1} - \tau_1$. This does not affect the order of our regret bound. With our assumption, because the all episode length is not bigger than the last episode, that is $t_{k+1} - 1 - (t_k + \tau_1) \le T_{K_T} - \tau_1$, then we can obtain,

$$\sum_{k=1}^{K_T}\sum_{t=t_k+\tau_1}^{t_{k+1}-1}\beta_k^i(z_t) \le \sum_{k=1}^{K_T}\sum_{t=1}^{T_{k_T}-\tau_1}\beta_k^i(z_t). \tag{30}$$

Note that $\sum_{s'\in\mathcal{S}}\left|P_*^i(s'\mid z_t) - \hat{P}_k^i(s'\mid z_t)\right| \le 2$ is always true. And with our assumption $\tau_1 \le \frac{T_1+K_T-1}{2}$, it is easy to show that when $N_{t_k}^i \ge T_{k_T} - \tau_1$, $\beta_k^i(z_t) \le 2$ holds. Then we can obtain,

$$\sum_{k=1}^{K_T} \sum_{t=1}^{T_{k_T}-\tau_1} \min\{2, \beta_k^i(z_t)\} \le \sum_{k=1}^{K_T} \sum_{t=1}^{T_{k_T}-\tau_1} 2\mathbb{I}(N_{t_k}^i < T_{k_T} - \tau_1)$$
$$+ \sum_{k=1}^{K_T} \sum_{t=1}^{T_{k_T}-\tau_1} \mathbb{I}(N_{t_k}^i \ge T_{k_T} - \tau_1) \sqrt{\frac{14S \log(2Nt_kT)}{\max(1, N_{t_k}^i(z_t))}}. \tag{31}$$

**Consider the first part in equation 31.** Obviously, the maximum of $N_{t_k}^i$ is $T_{k_T} - \tau_1$. Because there are totally $SA$ state-action pairs, therefore, the first part in equation equation 31 can be bounded as, $\sum_{k=1}^{K_T} \sum_{t=1}^{T_{k_T}-\tau_1} 2\mathbb{I}(N_{t_k}^i < T_{k_T} - \tau_1) \le 2(T_{k_T} - \tau_1)SA$. Due to $T_{k_T} = T_1 + K_T - 1$ and Lemma 10, we get ,

$$2(T_{k_T} - \tau_1)SA = 2(T_1 + K_T - \tau_1 - 1)SA = \mathcal{O}(\sqrt{T}).$$

**Consider the second part in 31.** Denote the $N_t^i(s, a)$ is the count of $(s, a)$ before time $t$(not including $t$). Due to we just consider the exploration phase in each episode, then $N_t^i(s, a)$ can be calculated as follows,

$$N_t^i(s, a) = \left| \{\tau < t, \tau \in [t_k, t_k + \tau_1], k \le k(t) : (s_\tau^i, a_\tau^i) = (s, a)\} \right|,$$

where $k(t)$ is the episode number where the time $t$ is in.

In the second part in equation 31, when $N_{t_k}^i \ge T_{k_T} - \tau_1$, based on our assumption $\tau_1 \le \frac{T_1 + K_T - 1}{2}$, we can get,

$$\tau_1 \le \frac{T_1 + K_T - 1}{2},$$
$$2\tau_1 \le T_1 + K_T - 1 = T_{k_T}.$$

therefore, $T_{k_T} - \tau_1 \ge \tau_1$. Because $N_{t_k}^i \ge T_{k_T} - \tau_1$, then $N_{t_k}^i(s, a) \ge \tau_1$. For any $t \in [t_k, t_k + \tau_1]$, we have

$$N_t^i(s, a) \le N_{t_k}^i(s, a) + \tau_1 \le 2N_{t_k}^i(s, a).$$

Therefore $N_t^i(s, a) \le 2N_{t_k}^i(s, a)$. Next we can bound the confidence set when $N_t(s, a) \le 2N_{t_k}(s, a)$ as follows,

$$\sum_{k=1}^{K_T} \sum_{t=1}^{T_{k_T}-\tau_1} \beta_k^i(z_t) \le \sum_{k=1}^{K_T} \sum_{t=t_k}^{t_{k+1}-1} \sqrt{\frac{14S \log(2Nt_kT)}{\max(1, N_{t_k}^i(z_t))}}$$
$$\le \sum_{k=1}^{K_T} \sum_{t=t_k}^{t_{k+1}-1} \sqrt{\frac{14S \log(2NT^2)}{\max(1, N_{t_k}^i(z_t))}}$$
$$= \sum_{t=1}^{T} \sqrt{\frac{28S \log(2NT^2)}{\max(1, N_t^i(z_t))}} \tag{32}$$
$$\le \sqrt{56S \log(2NT)} \sum_{t=1}^{T} \frac{1}{\sqrt{\max(1, N_t^i(z_t))}}.$$

where the second inequality in equation 32 is due to $t_k \le T$ for all episodes and the first equality is due to $N_t^i(s, a) \le 2N_{t_k}^i(s, a)$.

Then similar to Ouyang et al. (2017), since $N_t^i(z_t)$ is the count of visits to $z_t$, we have

$$\sum_{t=1}^{T} \frac{1}{\sqrt{\max(1, N_t^i(z_t))}} = \sum_z \sum_{t=1}^{T} \frac{\mathbb{I}_{\{z_t=z\}}}{\sqrt{\max(1, N_t^i(z))}}$$
$$= \sum_z \left( \mathbb{I}_{\{N_{T+1}^i(z)>0\}} + \sum_{j=1}^{N_{T+1}^i(z)-1} \frac{1}{\sqrt{j}} \right)$$
$$\le \sum_z \left( \mathbb{I}_{\{N_{T+1}^i(z)>0\}} + 2\sqrt{N_{T+1}^i(z)} \right) \le 3 \sum_z \sqrt{N_{T+1}^i(z)}.$$

Since $\sum_z N^i_{T+1}(z) \le T$, we have

$$3 \sum_z \sqrt{N^i_{T+1}(z)} \le 3 \sqrt{SN \sum_z N^i_{T+1}(z)} = 3\sqrt{SNT}. \tag{33}$$

With equation 32 and equation 33 we get

$$2H \sum_{k=1}^{K_T} \sum_{t=t_k}^{t_{k+1}-1} \beta^i_k(z_t) \le 6\sqrt{56} HS\sqrt{NT\log(NT)} \le 48HS\sqrt{NT\log(NT)}.$$

Then we can bound the equation 30 as follows,

$$\sum_{k=1}^{K_T} \sum_{t=t_k}^{t_{k+1}-1} \beta^i_k(z_t) \le 24S\sqrt{NT\log(NT)} + 2SA(T_1 + K_T - \tau_1 - 1). \tag{34}$$

Choose the $\delta = 1/T$ in Lemma 12, and based by Lemma 13, we obtain that

$$\mathbb{P}\left(P^i_k \notin B_k\right) = \mathbb{P}\left(P^i_* \notin B_k\right) \le \frac{1}{15Tt_k^6}.$$

Then we can obtain,

$$2\mathbb{E}_{\theta^*}\left[\sum_{k=1}^{K_T} T_k \left(\mathbb{I}_{\{\theta^* \notin B_k\}} + \mathbb{I}_{\{\theta_k \notin B_k\}}\right)\right] \le \frac{4}{15} \sum_{k=1}^{\infty} t_k^{-6} \le \frac{4}{15} \sum_{k=1}^{\infty} k^{-6} \le 1. \tag{35}$$

Therefore we obtain

$$2H\mathbb{E}_{\theta^*}\left[\sum_{k=1}^{K_T} T_k \left(\mathbb{I}_{\{\theta^* \notin B_k\}} + \mathbb{I}_{\{\theta_k \notin B_k\}}\right)\right] \le H. \tag{36}$$

Therefore, we can obtain the bound for one arm as follows,

$$\mathbb{E}_{\theta^*}\left[\sum_{k=1}^{K_T} \sum_{t=t_k+\tau_1+1}^{t_{k+1}-1} \left(\sum_{s' \in \mathcal{S}} \left(P^i_*(s' \mid z_t) - P^i_k(s' \mid z_t)\right) v(s', \theta_k)\right)\right] \tag{37}$$
$$\le H + 4SNH(T_1 + K_T - \tau_1 - 1) + 48HS\sqrt{NT\log(NT)}.$$

Next we consider the state transition of all arms. Recall that the states of all arms at time $t$ is $s_t$. Because every arm evolves independently, then the transition probability from state $s_t$ to state $s_{t+1}$ is as follows,

$$P(s_{t+1} \mid s_t, \theta^*) = \prod_{i=1}^{N} P^i_*(s^i_{t+1} \mid s^i_t),$$

where $P^i_*$ is the true transition functions of arm $i$. Based by the Lemma 11 and our assumption that all arms have the same state space $S$, we can obtain

$$\sum_{s_{t+1}} |P(s_{t+1} \mid s_t, \theta^*) - P(s_{t+1} \mid s_t, \theta_k)| \le \sum_i^N \left\|P^i_*(s^i_{t+1} \mid s^i_t) - P^i_k(s^i_{t+1} \mid s^i_t)\right\|_1 \tag{38}$$
$$\le N \left\|P^i_*(s^i_{t+1} \mid s^i_t) - P^i_k(s^i_{t+1} \mid s^i_t)\right\|_1.$$

Therefore, we can bound the $R_3^0$ as follows:

$$R_3^0 \le NH + 4SN^2H(T_1 + K_T - \tau_1 - 1) + 48SNH\sqrt{NT\log(NT)}. \tag{39}$$

**Bounding $R_3^1$.** Based on the Proposition 2, we know that

$$\left\|b^*_t - \hat{b}_t\right\|_1 \le L_1\|R^* - R_k\|_1 + L_2 \max_s \|P^*(s, :) - P_k(s, :)\|_2.$$

Note that the elements in the true transition matrix $P^*$ and the sampled matrix $P_k$ is between the interval $(0, 1)$. Then based on the facts about norm, we know that

$$\max_s \|P^*(s, :) - P_k(s, :)\|_2 \leq \|P^* - P_k\|_1.$$

Therefore , we can bound the belief error at any time as follows:

$$\left\| b_t^* - \hat{b}_t \right\|_1 \leq L_1 \|R^* - R_k\|_1 + L_2 \|P^* - P_k\|_1. \tag{40}$$

Recall in the confidence for $M_k$, the error bound is the same for $\|R^* - R_k\|_1$ and $\|P^* - P_k\|_1$, and based by the bound in equation 34 and equation 35, we can bound the $R_3^1$ as follows:

$$R_3^1 \leq H \mathbb{E}_{\theta^*} \left[ \sum_{k=1}^{K_T} \sum_{t=t_k}^{t_{k+1}-1} (L_1 \|R^* - R_k\|_1 + L_2 \|P^* - P_k\|_1) \right]$$

$$\leq (L_1 + L_2 N) H \mathbb{E}_{\theta^*} \left[ \sum_{k=1}^{K_T} \sum_{t=t_k}^{t_{k+1}-1} \left( 2\beta_k^i(z_t) + 2 \left( \mathbb{I}_{\{P^* \notin B_k\}} + \mathbb{I}_{\{P_k \notin B_k\}} \right) \right) \right] \tag{41}$$

$$\leq 48(L_1 + L_2 N) SH \sqrt{NT \log(NT)} + (L_1 + L_2 N) H$$
$$4(L_1 + L_2 N) SNH(T_1 + K_T - \tau_1 - 1).$$

**Bounding $R_3^2$.** Based on equation 34 and equation 35, we can bound $R_3^2$ as follows,

$$R_3^2 = S^2 H \mathbb{E}_{\theta^*} \left[ \sum_{k=1}^{K_T} \sum_{t=t_k+\tau_1+1}^{t_{k+1}-1} \|R^*(\cdot \mid s) - R_k(\cdot \mid s)\|_1 \right]$$

$$\leq S^2 H \mathbb{E}_{\theta^*} \left[ \sum_{k=1}^{K_T} \sum_{t=t_k+\tau_1+1}^{t_{k+1}-1} \left( 2\beta_k^i(z_t) + 2 \left( \mathbb{I}_{\{R^* \notin B_k\}} + \mathbb{I}_{\{R_k \notin B_k\}} \right) \right) \right] \tag{42}$$

$$\leq HS^2 + 4S^3 NH(T_1 + K_T - \tau_1 - 1) + 48HS^3 \sqrt{NT \log(NT)}.$$

Combine the bound in equation 39, equation 41 and equation 42, we bound the term $R_3$ as follows:

$$R_3 \leq 48(L_1 + L_2 N) SH \sqrt{NT \log(NT)} + 4(L_1 + L_2 N) SNH(T_1 + K_T - \tau_1 - 1)$$
$$+ (L_1 + L_2 N) H + NH + 4SN^2 H(T_1 + K_T - \tau_1 - 1) + 48SNH \sqrt{NT \log(NT)}$$
$$+ HS^2 + 4S^3 NH(T_1 + K_T - \tau_1 - 1) + 48HS^3 \sqrt{NT \log(NT)} \tag{43}$$
$$= 48(L_1 + L_2 N + N + S^2) SH \sqrt{NT \log(NT)} + (L_1 + L_2 N + N + S^2) H$$
$$+ 4(L_1 + L_2 N + N^2 + S^2) SNH(T_1 + K_T - \tau_1 - 1).$$

$\square$

### B.4 Bound $R_4$

**Lemma 9.** $R_4$ *satisfies the following bound*

$$R_4 \leq 48(L_1 + L_2 N + N + S^2) S r_{max} \sqrt{NT \log(NT)} + (L_1 + L_2 N + N + S^2) r_{max}$$
$$+ 4(L_1 + L_2 N + N + S^2) SA r_{max}(T_1 + K_T - \tau_1 - 1).$$

*Proof.* We can rewrite the $R_4$ as follows:

$$R_4 = \mathbb{E}_{\theta^*} \left[ \sum_{k=1}^{K_T} \sum_{t_k+\tau_1+1}^{t_{k+1}-1} \left( \sum_s r_k(s, a_t) \hat{b}_t(s) - \sum_s r^*(s, a_t) b_t^*(s) \right) \right]$$

$$\leq \mathbb{E}_{\theta^*} \left[ \sum_{t=1}^{T} \left( \sum_s r_k(s, a_t) \hat{b}_t(s) - \sum_s r_k(s, a_t) b_t^*(s) + \sum_s r_k(s, a_t) b_t^*(s) - \sum_s r^*(s, a_t) b_t^*(s) \right) \right] \tag{44}$$

where $r_k(s, a_t) = \sum_r r R_k^{a_t}(r \mid s)$ is the expect reward conditioned on the state $s$ of pulled arm and $a_t$, when the reward function is $R_k^{a_t}$. And $r^*(s, a_t) = \sum_r r R_*^{a_t}(r \mid s)$ is the expect reward conditioned on the state $s$ and $a_t$, with the true reward function $R_*^{a_t}$. The equation 44 is due to the add the term $\sum_s r_k(s, a_t) b_t^*(s)$ and subtract it.

Denote

$$R_4^0 = \mathbb{E}_{\theta^*}\left[\sum_{t=1}^T \left(\sum_s r_k(s, a_t)\hat{b}_t(s) - \sum_s r_k(s, a_t) b_t^*(s)\right)\right],$$

$$R_4^1 = \mathbb{E}_{\theta^*}\left[\sum_{t=1}^T \left(\sum_s r_k(s, a_t) b_t^*(s) - \sum_s r^*(s, a_t) b_t^*(s)\right)\right].$$

For $R_4^0$,

$$
\begin{aligned}
R_4^0 &= \mathbb{E}_{\theta^*}\left[\sum_{t=1}^T \left(\sum_s r_k(s, a_t)\hat{b}_t(s) - \sum_s r_k(s, a_t) b_t^*(s)\right)\right] \\
&= \mathbb{E}_{\theta^*}\left[\sum_{t=1}^T \left(\sum_s r_k(s, a_t)(\hat{b}_t(s) - b_t^*(s))\right)\right] \\
&\le r_{max}\mathbb{E}_{\theta^*}\left[\sum_{t=1}^T \left(\sum_s \left|\hat{b}_t(s) - b_t^*(s)\right|\right)\right]
\end{aligned}
\tag{45}
$$

where the last inequality is due to the fact $r_k(s, a_t) \le r_{max}$.

For $R_4^1$,

$$
\begin{aligned}
R_4^1 &= \mathbb{E}_{\theta^*}\left[\sum_{t=1}^T \left(\sum_s r_k(s, a_t) b_t^*(s) - \sum_s r^*(s, a_t) b_t^*(s)\right)\right] \\
&= \mathbb{E}_{\theta^*}\left[\sum_{t=1}^T \left(\sum_s [r_k(s, a_t) - r^*(s, a_t)] b_t^*(s)\right)\right] \\
&\le \mathbb{E}_{\theta^*}\left[\sum_{t=1}^T \left(\sum_s |r_k(s, a_t) - r^*(s, a_t)|\right)\right] \\
&\le \mathbb{E}_{\theta^*}\left[\sum_{t=1}^T \left(\sum_s \sum_r r\left|R_k^{a_t}(r \mid s) - R_*^{a_t}(r \mid s)\right|\right)\right] \\
&\le S r_{max}\mathbb{E}_{\theta^*}\left[\sum_{t=1}^T \left(\|R_k^{a_t} - R_*^{a_t}\|_1\right)\right]
\end{aligned}
\tag{46}
$$

where the first inequality in 46 is due to $b_t^*(s) \le 1$, $r_k(s, a_t) - r^*(s, a_t) \le |r_k(s, a_t) - r^*(s, a_t)|$ and the second inequality is due to $\sum_r [R_k^{a_t}(r \mid s) - R_*^{a_t}(r \mid s)] \le \|R_k^{a_t} - R_*^{a_t}\|_1$.

Based on the equation 41, we can bound the $R_4^0$,

$$
\begin{aligned}
R_4^0 \le{}& 48(L_1 + L_2 N)S r_{max}\sqrt{NT \log(NT)} + (L_1 + L_2 N)r_{max} \\
&+ 4(L_1 + L_2 N)SN r_{max}(T_1 + K_T - \tau_1 - 1).
\end{aligned}
$$

Note that for any reward function $R(r \mid z)$ in confidence set $\mathcal{M}_k$, the reward function satisfies,

$$\sum_{r \in \mathcal{R}}\left|R(r \mid z) - \hat{R}_k^i(r \mid z)\right| \le \beta_k^i(z)$$

Then based on equation 42, we get

$$R_4^1 \le 48 S^2 r_{max}\sqrt{NT \log(NT)} + 2S^2 N r_{max}(T_1 + K_T - \tau_1 - 1) + S r_{max}.$$

Then we can obtain the final bound:

$$
\begin{aligned}
R_4 &\leq 48(L_1 + L_2N + S)Sr_{max}\sqrt{NT\log(NT)} + 4(L_1 + L_2N + S)SNr_{max}(T_1 + K_T - \tau_1 - 1) \\
&\quad + (L_1 + L_2N + S)r_{max} \\
&\leq 48(L_1 + L_2N + N + S^2)Sr_{max}\sqrt{NT\log(NT)} + (L_1 + L_2N + N + S^2)r_{max} \\
&\quad + 4(L_1 + L_2N + N + S^2)SNr_{max}(T_1 + K_T - \tau_1 - 1)
\end{aligned}
$$

where the last inequality is due to $S \leq N + S^2$. $\qquad\square$

### B.5 THE TOTAL REGRET

Next we bound the episode number.

**Lemma 10.** *(Bound the episode number) With the convention $T_1 = \left\lceil \frac{\sqrt{T}+1}{2} \right\rceil$ and $T_k = T_{k-1} + 1$, the episode number is bounded by $K_T = \mathcal{O}(\sqrt{T})$.*

*Proof.* Note that the total horizon is $T$. The length of episode $k$ is $T_k = T_1 + k - 1$. Then we can get,

$$
\begin{aligned}
T &= T_1 + T_2 + ... + T_{k_T} \\
&= T_1 + (T_1 + 1) + ... + (T_1 + K_T - 1) \\
&= K_T T_1 + (1 + 2 + ... + K_T - 1) \\
&= K_T T_1 + \frac{K_T(K_T - 1)}{2}.
\end{aligned}
\tag{47}
$$

Therefore,

$$
K_T^2 + (2T_1 - 1)K_T - 2T = 0.
\tag{48}
$$

With the convention $T_1 = \left\lceil \frac{\sqrt{T}+1}{2} \right\rceil$, then we can get $K_T = \mathcal{O}(\sqrt{T})$ $\qquad\square$

Denote $C_1 = L_1 + L_2N + N^2 + S^2$, $C_2 = H + r_{max}$ and $C_3 = T_1 + K_T - \tau_1 - 1$, then we can get the final regret:

$$
\begin{aligned}
R_T &= \text{Regret(A)} + R_1 + R_2 + R_3 + R_4 \\
&\leq \tau_1 \Delta R K_T + H K_T + 48C_1 SH\sqrt{NT\log(NT)} + 4C_1C_3SAH + C_1H \\
&\quad + 48C_1 Sr_{max}\sqrt{NT\log(NT)} + 4C_1C_3SAr_{max} + C_1r_{max} \\
&\leq (\tau_1\Delta R + H)\sqrt{T} + 48C_1S(H + r_{max})\sqrt{NT\log(NT)} \\
&\quad + 4C_1SA(r_{max} + H)\sqrt{T} + C_1(H + r_{max}) \\
&= 48C_1C_2S\sqrt{NT\log(NT)} + (\tau_1\Delta R + H + 4C_1C_2SN)\sqrt{T} + C_1C_2.
\end{aligned}
$$

Thus, we get the final Theorem.

**Theorem 2.** *Suppose Assumptions 1,2 hold and the Oracle returns the optimal policy in each episode. The Bayesian regret of our algorithm satisfies*

$$
R_T \leq 48C_1C_2S\sqrt{NT\log(NT)} + (\tau_1\Delta R + H + 4C_1C_2SN)\sqrt{T} + C_1C_2,
$$

*where $C_1 = L_1 + L_2N + N^2 + S^2$, $C_2 = r_{max} + H$ are constants independent with time horizon $T$, $L_1 = \frac{4(1-\epsilon_1)^2}{N\epsilon_1^2\epsilon_2}$, $L_2 = \frac{4(1-\epsilon_1)^2}{\epsilon_1^3}$, $\epsilon_1$ and $\epsilon_2$ are the minimum elements of the functions $P^*$ and $R^*$, respectively. $\tau_1$ is the fixed exploration length in each episode, $\Delta R$ is the biggest gap of the reward obtained at each two different time, $H$ is the bounded span, $r_{max}$ is the maximum reward obtain each time, $N$ is the number of arms and $S$ is the state size for each arm.*

**Remark 3.** *(Approximation error.) If the oracle returns an $\epsilon_k$-approximate policy $\tilde{\pi}_k$ in each episode instead of the optimal policy. That is to say, $r(b, \tilde{\pi}_k(b)) + \sum_r P(r \mid b, \tilde{\pi}_k(b), \theta) v(b', \theta) \leq \max_a \{ r(b, a) + \sum_r P(r \mid b, a, \theta) v(b', \theta) \} - \epsilon_k$. Then we should consider the extra regret $\mathbb{E}\left[ \sum_{k:t_k \leq T} (T_k - \tau_1) \epsilon_k \right]$ in exploitation phase. If we control the error as $\epsilon_k \leq \frac{1}{T_k - \tau_1}$, then we can bound the extra regret as $\mathbb{E}\left[ \sum_{k:t_k \leq T} (T_k - \tau_1) \epsilon_k \right] \leq k_T = \mathcal{O}(\sqrt{T})$ (Lemma 10). Thus the approximation error in the computation of optimal policy is only additive to the regret of our algorithm.*

$\square$

## C  POSTERIOR DISTRIBUTION

Note that we assume the state transition is independent of the action for each arm. Denote the states visited history from time $0$ till $t$ of arm $i$ as $s_{0:t}^i$ and the reward collected history is $r_{0:t}^i$. And the action history from time $0$ to $t$ is $a_{0:t}^i$. Denote $N_{s,s'}^i(s_{0:t}^i)$ as the occurence time of state evolves from $s$ to $s'$ for arm $i$ in the state history $s_{0:t}^i$. Hence, if the prior $g(P_i(s, \cdot))$ is Dirichlet $(\phi_{s,s_1}^i, \ldots, \phi_{s,\mathcal{S}_i}^i)$, then after the observation of history $s_{0:t}^i$, the posterior $g(P_i(s, \cdot) \mid s_{0:t}^i)$ is Dirichlet $(\phi_{s,s_1}^i + N_{s,s_1}^i(s_{0:t}^i), \ldots, \phi_{s,\mathcal{S}_i}^i + N_{s,\mathcal{S}_i}^i(s_{0:t}^i))$ (Ross et al., 2011).

Similarly, if the prior $g(R_i(s, \cdot))$ is Dirichlet $(\psi_{s,r_1}^i, \ldots, \psi_{s,r_k}^i)$, then after the observation of reward history $r_{0:t}^i$ and $s_{0:t}^i$, the posterior $g(R_i(s, \cdot) \mid r_{0:t}^i, s_{0:t}^i)$ is Dirichlet $(\psi_{s,r_1}^i + N_{s,r_1}^i(s_{0:t}^i, r_{0:t}^i), \ldots, \psi_{s,r_k}^i + N_{s,r_k}^i(s_{0:t}^i, r_{0:t}^i))$, and $N_{s,r}^i$ is the number of times the observation $(s, r)$ appears in the history $(s_{0:t}^i, r_{0:t}^i)$.

Here we drop the arm index and consider a fixed arm. For the unknown transition function, we assume its prior $g_0(P^i) = f(\frac{P^i - \epsilon_1 \mathbf{1}}{1 - \epsilon_1} \mid \phi^i)$. We consider this special prior is due to the minimum elements of the transition matrix is bigger than $\epsilon_1$. Next we show the details that how to update the posterior distribution for unknown $P$ and omit the details of unknown reward function $R$.

$$
\begin{aligned}
g(P \mid a_{0:t-1}, r_{0:t-1}) &= \frac{P(r_{0:t-1}, s_t \mid P, a_{0:t-1}) g(P, a_{0:t-1})}{\int P(r_{0:t-1}, s_t \mid P, a_{0:t-1}) g(P, a_{0:t-1}) \, dP} \\
&= \frac{\sum_{s_{0:t-1} \in S^t} P(r_{0:t-1}, s_{0:t} \mid P, a_{0:t-1}) g(P)}{\int P(r_{0:t-1}, s_t \mid P, a_{0:t-1}) g(P, a_{0:t-1}) \, dP} \\
&= \frac{\sum_{s_{0:t-1} \in S^t} g(P) \prod_{i=1}^t P(s_i \mid s_{i-1})}{\int P(r_{0:t-1}, s_t \mid P, a_{0:t-1}) g(P, a_{0:t-1}) \, dP} \\
&= \frac{\sum_{s_{0:t-1} \in S^t} g(P) \left[ \prod_{s,s'} (\frac{P(s' \mid s) - \epsilon_1}{1 - \epsilon_1})^{N_{ss'}(s_{0:t})} \right]}{\int P(r_{0:t-1}, s_t \mid P, a_{0:t-1}) g(P, a_{0:t-1}) \, dP}.
\end{aligned}
$$

where the last equality is due to the prior for unknown $P^i$ is $g_0(P^i) = f(\frac{P^i - \epsilon_1 \mathbf{1}}{1 - \epsilon_1} \mid \phi^i)$.

Next we show the Bayesian approach to learning unknown $P$ and $R$ with the history $(a_{0:t-1}, r_{0:t})$. Since the current state $s_t$ of the agent at time $t$ is unobserved, we consider a joint posterior $g(s_t, P, R \mid a_{0:t-1}, r_{0:t})$ over $s_t, P$, and $R$ (Ross et al., 2011). The most parts are similar to Ross

et al. (2011), except for our special priors.

$$
\begin{aligned}
g\left(s_t, P, R \mid a_{0:t-1}, r_{0:t-1}\right) &\propto P\left(r_{0:t}, s_t \mid P, R, a_{0:t-1}\right) g\left(P, R, a_{0:t-1}\right) \\
&\propto \sum_{s_{0:t-1} \in S^t} P\left(r_{0:t}, s_{0:t} \mid P, R, a_{0:t-1}\right) g(P, R) \\
&\propto \sum_{s_{0:t-1} \in S^t} g\left(s_0, P, R\right) \prod_{i=1}^{t} P(s_i \mid s_{i-1}) R(r_i \mid s_i) \\
&\propto \sum_{s_{0:t-1} \in S^t} g\left(s_0, P, R\right) \left[ \prod_{s,s'} (\frac{P(s' \mid s) - \epsilon_1}{1 - \epsilon_1})^{N_{ss'}(s_{0:t})} \right] \times \\
&\qquad \left[ \prod_{s,r} (\frac{R(r \mid s) - \epsilon_2}{1 - \epsilon_2})^{N_{sr}(s_{0:t}, r_{0:t-1})} \right]
\end{aligned}
$$

where $g\left(s_0, P, R\right)$ is the joint prior over the initial state $s_0$, transition function $P$, and reward function $R$; $N_{ss'}\left(s_{0:t}\right)$ is the number of times the transition $(s, s')$ appears in the history of state-action $(s_{0:t})$; and $N_{sr}\left(s_{0:t}, r_{0:t-1}\right)$ is the number of times the observation $(s, r)$ appears in the history of state-rewards $(s_{0:t}, r_{0:t-1})$.

## D   TECHNICAL RESULTS

**Proposition 1.** *(Uniform bound on the bias span (Zhou et al., 2021)). If the belief MDP satisfies Assumption 1,2, then for $(J(\theta), v(:, \theta))$ satisfying the Bellman equation (2), we have the span of the bias function $\mathrm{span}(v, \theta) := \max_\theta \max_b v(b, \theta) - \min_{b, \theta \in \mathcal{B}} v(b, \theta)$ is bounded by $H(\epsilon)$, where*

$$
H(\epsilon) := \frac{8\left(\frac{2}{(1-\alpha)^2} + (1+\alpha)\log_\alpha \frac{1-\alpha}{8}\right)}{1-\alpha}, \quad with \; \alpha = \frac{1-\epsilon}{1-\epsilon/2} \in (0,1)
$$

It is easy to check that $H(\alpha)$ is increasing with $\alpha$. Since $\alpha$ is decreasing with $\epsilon$ and we assume the smallest element in transitions matrix is $\epsilon_1$, the span function can be bounded by $H(\epsilon_1)$.

**Proposition 2.** *(Controlling the belief error (Xiong et al., 2022c)). Suppose Assumption 1,2 hold. Given $(R_k, P_k)$, an estimator of the true model parameters $(R^*, P^*)$. For an arbitrary reward-action sequence $\bar{r}_t, \bar{a}_t$, let $\hat{b}_t(\cdot, R_k, P_k)$ and $b_t(\cdot, R^*, P^*)$ be the corresponding beliefs in period t under $(R_k, P_k)$ and $(R^*, P^*)$ respectively. Then there exists constants $L_1, L_2$ such that*

$$
\left\| b_t(\cdot, R^*, P^*) - \hat{b}_t(\cdot, R_k, P_k) \right\|_1 \leq L_1 \|R_k - R^*\|_1 + L_2 \max_s \|P^*(m, :) - P_k(m, :)\|_2,
$$

*where $L_1 = \frac{4(1-\epsilon_1)^2}{N\epsilon_1^2 \epsilon_2}, L_2 = \frac{4(1-\epsilon_1)^2}{\epsilon_1^3}$, $\epsilon_1$ and $\epsilon_2$ are the minimum elements of the functions $P^*$ and $R^*$, respectively.*

**Lemma 11.** *(Lemma 13 in Jung et al. (2019)) Suppose $a_k$ and $b_k$ are probability distributions over a set $[n_k]$ for $k \in [K]$. Then we have*

$$
\sum_{x \in \otimes_{k=1}^K [n_k]} \left| \prod_{k=1}^K a_{k, x_k} - \prod_{k=1}^K b_{k, x_k} \right| \leq \sum_{k=1}^K \|a_k - b_k\|_1.
$$

**Lemma 12.** *(Lemma 17 in Auer et al. (2008)) For any $t \geq 1$, the probability that the true MDP $M$ is not contained in the set of plausible MDPs $\mathcal{M}(t)$ at time t is at most $\frac{\delta}{15t^6}$, that is*

$$
\mathbb{P}\{M \notin \mathcal{M}(t)\} < \frac{\delta}{15t^6}.
$$

**Lemma 13.** *(Posterior Sampling (Ouyang et al., 2017)). In TSEETC, $t_k$ is an almost surely finite $\sigma\left(\mathcal{H}_{t_k}\right)$-stopping time. If the prior distribution $g_0(P), g_0(R)$ is the distribution of $\theta^*$, then for any measurable function g,*

$$
\mathbb{E}\left[g\left(\theta^*\right) \mid \mathcal{H}_{t_k}\right] = \mathbb{E}\left[g\left(\theta_k\right) \mid \mathcal{H}_{t_k}\right].
$$

