# OpenReview forum: "ONLINE RESTLESS BANDITS WITH UNOBSERVED STATES"
_ICLR.cc/2023/Conference — Submitted to ICLR 2023_

### Official Review · Reviewer_EDfN · 2022-10-24

**Confidence:** 4
**Correctness:** 4
**Technical Novelty And Significance:** 3
**Empirical Novelty And Significance:** Not applicable
**Recommendation:** 6

**Clarity, Quality, Novelty And Reproducibility:**

The paper has a few English typos at different places which sometimes makes the mathematical difficult to parse. The setting studied in the paper is quite classical.  The novelty is harder to judge for me (see my comment in the "weaknesses" above) but the method and algorithm proposed seems quite classical.

I could not find the code to check reproducibility.

**Strength And Weaknesses:**

I like the problem studied in this paper. This particular restless bandit setting has been studied in the literature and can represent many problems one to send probes to estimate communication channels' quality. The regret bound is not particularly new but improves over previous work.

Weaknesses:

My main problem with this paper is that the contributions are hidden: the authors explain their results but their is little comment about their relevance or their originality. For instance: the main claim of the authors is that the algorithm has a O(\sqrt{t}) Bayesian regret. To me, it seems that any classical learning algorithm will have a linear regret for this problem (the only difficulty seems the unbounded state space if an arm is not activated for a long time). Hence: what makes this algorithm interesting and what is this specific form of explore-and-commit with multiple episodes? Is this specific to this particular restless bandits or could/should it be used elsewhere?

Also, Assumption 1 seems quite strong. In particular, it is not satisfied by the "Dirichlet prior" studied in the paper. It seems that this diminishes the value of the proposed algorithm because the theoretical properties are not applicable to the algorithm studied.

Minor comments:
- The paper needs an oracle to compute the policy.
- page 4: in general, for weakly communicating MDPs, there are many functions satisfying (2) (not just up to an additive constant). Hence the span is not clearly defined.
- On Figures 6: the authors plot the log of the performance as a function of the log of time -> using a log-log plot with proper scaling would be easier to read.
- Is not Lemma 1 obvious?


**Summary Of The Paper:**

This paper proposes an algorithm, called TSEETC that aims at providing a low-regret learning algorithm for partially observable multi-armed bandits. In this paper, the decision maker faces N Markov reward processes and chooses which arm to activate at each time instant. The state of a given arm is revealed to the decision maker only when this arm is activated. As is classically done in the literature, by changing the state space, this Markovian bandit problem with non-observable states is transformed in a *restless* Markovian bandit problem with *observable* state.

The authors assume that the decision maker does not know the transition probabilities nor reward and design a learning algorithm for this setting. Under a quite strong condition that all states can ve attained with probability \varepsilon, they prove that this algorithm has a $O(\sqrt{T})$ regret.



**Summary Of The Review:**

This paper uses a restless bandit approach to solve a Markovian bandit problem where states are only observable upon activating an arm. The authors derive an algorithm that has a good regret bound. The result is new but the novelty of the method is unclear to me.


After reading the rebuttal, the contributions are clearer to me.

---

> ### Author Response · Authors · 2022-11-19
> **The hidden contribution and  discussion about Assumption 1**
>
> Thank you for your valuable comments, especially about clarifying the contribution and the discussion about Assumption 1. First of all, We believe there is a major misunderstanding of our work. As we emphasized in the introduction and problem setting, the states of arms are never observed, not even after pulling, which makes the problem much more difficult than the case where the state of the pulled arm is observed. In fact, one of the main challenges in our work is to handle the unobserved states.
> We address your concerns accordingly and upload the revised version.
>
> R1：The contribution is hidden.
>
> Q1：Actually classical algorithms such as ETC do not always suffer a linear regret. Zhou et al considers a similar problem. Their algorithm uses UCB for the exploration phase of episodic ETC and achieves a regret bound of $\tilde{\mathcal{O}}(T^{2/3})$.  Our TSEETC uses Thompson sampling instead in the exploration phase to obtain a better estimation bound for the average error as in Lemma 2 and hence a smaller regret, but only in the Bayesian sense.
> Specially, we update the posterior distributions about unknown parameters as the mixture of each combined distribution and propose the well-controlled episode length increasing manner to guarantee that the episode number is upper bounded by $\sqrt{T}$.
>
>
> R2：Assumption 1 seems quite strong.
>
> Q2：We agree that our assumption is strong. However, this problem is quite difficult, so we make assumption 1 for tractability as in Zhou et al. It would be interesting to see whether we can relax this assumption in future work. We thank the reviewer for pointing out the gap between assumption 1 and the Dirichlet prior. We have modified the prior  to enforce assumption 1.
> More precisely, the revised prior for $P^i$ is $f(\frac{P^i - \epsilon_1 \mathbf{1}}{1-S\epsilon_1} \mid \phi^i)$, where $f(\cdot \mid \phi^i)$ is the Dirichlet distribution, i.e. $P^i = \epsilon_1 \mathbf{1} + (1-S\epsilon_1) \tilde P^i$, where $\tilde P_i$ has a Dirichlet distribution. The algorithm and the analysis only require minor modifications, which we incorporated in the revised version.
>
>
> R3：an oracle to compute the policy
>
> Q3：The introduction of an oracle is not uncommon in the restless bandit problem, such as in Wang et al and Zhou et al. In fact, various approximation methods with accuracy guatrantee are proposed the solve the POMDP bellman equation.
> And as we have explained in Remark 3 in Page 7, the approximation error does not influence the order of regret.
>
>
> R4：the span is not clearly defined
>
> Q4：To be honest, we don't quite understand this problem.
> The span function is defined as the difference between the maximal value function $v_{max}(b,\theta )$ and the minimum
> $v_{min}(b,\theta )$. Since the value function is finite in our setting, the span function exists.
> And we give the explicit bound of span function in the newly added Proposition 2.
>
>
> R5：log-log plot for figure 2
>
> Q5：We have revised the figure to make its meaning clearer.
>
>
> R6：Is not Lemma 1 obvious?
>
> Q6：Yes, Lemma 1 is indeed intuitive.

---

### Official Review · Reviewer_csbH · 2022-10-24

**Confidence:** 3
**Correctness:** 4
**Technical Novelty And Significance:** 2
**Empirical Novelty And Significance:** Not applicable
**Recommendation:** 5

**Clarity, Quality, Novelty And Reproducibility:**

- It appears that the main challenge tackled in this paper is the fact that the states are unobserved. The authors successfully overcome this challenge via an adequate bayesian approach. However, the technical novelty of how they tackle this challenge is somehwat illusive in the main paper and I believe it deserves more explanation. It would also improve the readability of their proofs if the authors add to section 4.1 and 4.2, theoretical guarantees on the performance of their belief update procedure.
-  I am struggling to find an insight in achieving $\sqrt{T}$? In a sense, the provided result confirms that restless bandits with  unobserved states is as easy restless bandits with observed states (since both yield regret of order $\sqrt{T}$). But perhaps there are some subltelties to be discussed. Can the authors add some comments on this?
- The authors didn't provide any discussion on the frequentist regret. Can the authors explain whether they can obtain similar guarantees for the frequentist regret of their algorithm?
-  Is Lemma 2 worth adding in the main? shouldn't be in the appendix?
- The authors mention that colored-UCRL2 of Ortner et al.(2012) is computationally demanding in comparison with Thomson based sampling. What about the oracle used to compute the optimal policy of a POMDP? can that be also computationally demanding?
- The use of an explore-than-commit or episodes is somewhat standard in the literature. Is there something fundamentally important about using this in this setting here? If so please motivate.


**Strength And Weaknesses:**

Weaknesses

 -  Assumption 1 seems to be too strong in comparison with a weakly communicating MDP, could the authors motivate why they restrict themselves to this assumption.
-  (Literature review) I believe the paper: "the restless hidden markov bandit with linear rewards and side information" (see arxiv version https://arxiv.org/pdf/1910.10271.pdf ) is extremely relevant. The author did not cite this work nor discussed it. It seems an instance dependent regret bound of order $\log(T)$ is provided. How does this result compare with $\sqrt{T}$ bayesian regret bound provided by the authors here? A thorough comparison is needed here!
- The readability of the paper can be improved and a clear explanation clarifying how challenging is the task of tackling unseen states is somewhat illusive in the main first 8 pages of the paper. I think the authors should focus on this aspect a bit more.

Strengths

+ The setting of restless bandits with unobserved states appear to be understudied to the best of my knowledge.
+ Although I haven't thoroughly read most of the technical proofs, the results appear sound and technically correct.



**Summary Of The Paper:**

The authors consider a restless bandit problem where the states are unobserved. The authors propose algorithm Thompson Sampling with an Episodic Explore-Then-Commit (TSEETC). In the exploration phase, the algorithm uses a bayesian approach, based on mixtures of Dirichelet priors to update the unkown parameters and beliefs. In the the explotation phase the algorithm simply commits the estimated model and selects actions greedily.  This process is repeated for each episode. The authros porivde a bayesian regret guarantee of TSEETC scaling as $\sqrt{T}$.

**Summary Of The Review:**

My current recommendation of the paper is a score of 5 but I lean towards rejection, because of the following key points:

- The study of restless bandit with unobserved states is appreciated, the proposed algorithm with a guarantee seems sound, but a relevant paper is not discussed (see https://arxiv.org/pdf/1910.10271.pdf ).
- The major ingredient of TEECES to accommodate for unobserved states is the belief estimation part. More explanation and theoretical insight is needed about this part.
- The design of TEECES relies on episodic learning and an explore-than-commit strategy. These are standard in the literature.
- The readability of the paper can be improved

More questions on these points above. I may change my recommendation towards either acceptance or rejection depending on the authors response.

---

> ### Author Response · Authors · 2022-11-19
> **A relevant paper and the unobserved states**
>
> Thank you for your valuable comments, especially about the paper we do not discussed and the unseen states. We address your concerns accordingly and upload the rebuttal revision.
>
> Q1: The motivation for Assumption 1
>
> R1: We agree that our assumption is much stronger than the weakly communicating assumption and the latter is more desirable. However, this problem is quite difficult, so we make assumption 1 for tractability as in Zhou et al. It would be interesting to see whether we can relax this assumption in future work.
>
> Q2: comparison  with the paper “the restless hidden markov bandit with linear rewards and side information”
>
> R2: Thanks for pointing out this missing reference. We have added a discussion in the revised version. A key difference in the problem settings is that their reward function has a linear structure, which is not assumed in our setting. As pointed by the authors, this linear structure is quite a bit of side information that the decision maker can take advantage of for decision making. Another difference is that their $\log(T)$  regret bound is  instance-dependent, while our Bayesian bound $\tilde{\mathcal{O}}(\sqrt{T})$
> is instance-independent. It's typically the case, e.g. for UCB, that instance-dependent bounds are tighter than instance-independent bounds, although their bound is not directly comparable to ours due to the aforementioned difference in problem settings.
>
> Q3: tackling unseen states deserves more explanation.
>
> R3: Thanks for the suggestions. We add more explanations about the unknown states in introduction and section 4.
> The theoretical guarantee about the belief update is shown in Proposition 2.
> We show the estimation errors about unknown parameters is bounded by $\sqrt{T}$(newly added Lemma 2), which helps us obtain the final regret bound $\tilde{\mathcal{O}}(\sqrt{T})$.
>
> Q4: Intuition behind the regret bound $\tilde{\mathcal{O}}(\sqrt{T})$ and is restless bandits with unobserved states as easy restless bandits with observed states ?
>
> R4:  Compared with the state-of-the-art algorithm based on spectral estimator and UCB (Zhou et al), TS gives a better estimation bound for the average error as  in Lemma 2 and hence a smaller regret, but only in the Bayesian sense.
> Actually it is unclear yet  whether restless bandits with unobserved states is as easy as restless bandits with observed states.
> Our regret bound of $\tilde{\mathcal{O}}(\sqrt{T})$ is in the Bayesian sense. The current best frequentist regret bound (Zhou et al) for the unobserved state case is $\tilde{\mathcal{O}}(T^{2/3})$, which does not yet match the $\tilde{\mathcal{O}}(\sqrt{T})$ bound for observed state case.
>
>
> Q5: discussion on the frequentist regret
>
> R5: Unfortunately, we do not yet know whether our algorithm TSEETC can achieve $\sqrt{T}$ frequentist regret. It would be interesting to investigate this in the future.
>
>
> Q6: position of Lemma 2
>
> R6: Thanks for your suggestion. We moved Lemma 2 to the appendix in the revised version.
>
> Q7: computational complexity of the oracle
>
> R7: We apologize for this confusion. Our previous statement was not very accurate and we have revised it.
> As for computational complexity,
> Our TSEETC algorithm and color-UCRL2 both need to solve the Bellman equation and the difference is that
> we update posterior as mixtures and they should search for the optimistic model in confidence region. In particular, the optimal policy of a POMDP can be  approximated with the accuracy guarantee and low computational complexity.
> Though the candidate parameters in confidence region can be reduced, each candidate parameter needs to query the oracle to solve bellman equation. In this sense, it is easier to approximate
> TSEETC than color-UCRL2.
>
> Q8: motivation for ETC algorithm
>
> R8: We are not sure whether there is anything that mandates an ETC style algorithm, but it is kind of natural to estimate the unknown parameters first, which matches the ETC framework. We also considered algorithms without episodes, but did not work out.

---

### Official Review · Reviewer_WLZD · 2022-10-27

**Confidence:** 3
**Correctness:** 4
**Technical Novelty And Significance:** 3
**Empirical Novelty And Significance:** Not applicable
**Recommendation:** 6

**Clarity, Quality, Novelty And Reproducibility:**

The paper is clear and easy to read. The novelty and comparison against existing literature is clear and seems comprehensive.

**Strength And Weaknesses:**

## Strengths

– I think the key strength of the paper lies in establishing the O(\sqrt(T)) regret bound, which as the paper describes is an improvement over known bounds in the literature for RMABs. However, I think this positive also comes with the caveat that the RMAB considered here is rather simple with each arm representing a Markov Chain rather than an MDP (more on this under “weaknesses”)

– The paper is very well written and is pleasurable to read. The coverage of existing related work is excellent and I believe the paper does a great job of pointing out limitations of previous work and the new contributions.

## Weaknesses

– Model: Markov Chains
While the paper does a good job of highlighting the limitations of previous work and how all the previously known bounds are weaker than the one proposed, I think a key factor is that the setting considered makes it considerably simple: the RMAB arms are Markov chains, whose state transition probabilities are unaffected by actions taken. In most literature on RMABs, each arm is typically modeled as an MDP, which makes action planning considerably difficult.

– The empirical experiments could compare against more interesting baselines: For example, the paper mentions the Liu et. al. 2010 paper in related work saying how their log(T) regret doesn’t mean much. However, their setting seems most relevant as they also consider RMAB arms with Markov chain. I’d be interested in checking how their algorithm compares against the proposed algorithm.

– Although generally well written, the paper has several typos/missing words which need cleaning up (for eg: Page 2 – exiting, show that outperforms, Page 3 – methods to unknown states, reward functions)

**Summary Of The Paper:**

This paper considers the online Restless Bandit (RMAB) problem, where each arm of the RMAB is a Markov chain. The state space of each arm can be potentially unique, the states are assumed to be unobserved and the reward function is unknown. The goal is to design efficient algorithms that determine which arm to pull each round, so as to minimize the accumulated regret.


This paper proposes a Thompson Sampling based learning algorithm called TSEETC, which operates using alternating explore and exploit stages within each episode. The key contribution of the paper lies in establishing a Bayesian regret bound of O(\sqrt(T)). Finally the paper also presents proof-of-concept empirical experiments to corroborate the theoretical results.


**Summary Of The Review:**

Theoretically grounded paper, with good theoretical contribution improving over previous work, generally well-written

---

> ### Author Response · Authors · 2022-11-19
> **Action-independent state transitions and the relevant baseline should consider**
>
> Thank you for your valuable comments, especially about the action-independent state transitions and the relevant baseline we should consider. We address your concerns accordingly and upload the revised version.
>
> Q1：The state transition is independent of the action.
>
> R1：We agree that many works on RMAB  allow the state transition to depend on the action, which is more difficult than action-independent transitions. However, this does not mean their settings are more difficult than ours, as they assume the states are observable, either all the time or after pulling, which makes it much easier to estimate the unknown parameters. In contrast, we consider the setting that the states are never observed, not even after pulling. In this case, it is challenging to handle the unknown states and control the estimation error. In fact, even when the transitions are action-independent, the state-of-the-art algorithm (Xiang Zhou et.al.) has a regret of $\mathcal{O}(T^{2/3})$, worse than the $\mathcal{O}(\sqrt{T})$ bound for the observable state setting.
> Our algorithm improves the bound for the unobserved state case to $\mathcal{O}(\sqrt{T})$, although only in the Bayesian sense. It is not yet known whether this bound holds for frequentist regret in the action-independent setting. The  action-dependent setting would be  more difficult and require further investigation.
>
> Q2：The empirical experiments should consider the baseline in Liu et. al. 2010.
>
> R2:   Thanks for this suggestion. We added the baseline  RUCB (Liu et. al. 2010) in our experiment and the results are shown in Figures 1 and 2.  RUCB has linear regret, which is not surprising, as our definition of regret (equation 3) uses a stronger oracle than that considered in  RUCB.
>
> Q3:  several typos/missing words
>
> R3:  Thanks for pointing out this problem. We have corrected the errors and carefully check the other parts of our paper to make its meaning clearer.

---

### Official Review · Reviewer_PrWt · 2022-11-03

**Confidence:** 4
**Correctness:** 3
**Technical Novelty And Significance:** 4
**Empirical Novelty And Significance:** 3
**Recommendation:** 6

**Clarity, Quality, Novelty And Reproducibility:**

Clarity in writing could be improved. Quality and ovelty have been evaluated in **Strength/Weakness/Concern** in detail. Overall, this paper has sufficient novelty for the probelm it studies and the results it claims to get, which I may need more evidence/intuitions to be convinced. If the notation could be revised carefully throughout the paper, then the quality of presentation is good. I didn’t check the reproducibility of simulations but I’d like to believe the results are reproducible.


**Strength And Weaknesses:**

**Strengths**
1. This paper is a solid work providing theoretical understanding for a well defined problem, which has characteristics from both bandit and POMDP and is novel with no existing work addressing the exact same setting.
2. The $O(\sqrt{T})$ dependency, matching the lower bound dependency on $T$, is a significant improvement compared to the existing bounds of $T^{2/3}$. However, I’m not fully convinced by this improved dependency and have concern on the regret’s dependency on S and N, the number of states and arms respectively. See more detailed discuss in **weaknesses** and **concerns**.

**Weaknesses**
1. **Notations are somewhat sloppy**: To name a few which cause me the most trouble while reading:
     - Even in the main theorem **Theorem 1**: the notation $A$ comes out of nowhere, I assume it should be the number of arms $N$.
     - In the main algorithm **Algorithm 2**:
            (i) Line 4, $g_{t_k} (P)$ and $g_{t_k} (R)$ could be misleading. If following **Lemma 1**, $g_{t_k} (P)$ should refer to the posterior of $P$ conditoned on the history up to $t_k$, however it could also mean $g_{t_{k-1} + \tau_1}(P)$, which is what I assume the auther is actually referring to. This two interpretation have drastic difference since it depends on whether the data from the exploitation phase is used to update the posterior or not.
             (ii) Line 12, it's no clear what are the obtained $\bar r_{\tau_1}$ and $\bar b_{\tau_1}$, though for this case I can guess them from the context.
     - Some others in the main text like $M^*$, $M_k$ on page 9. Also I came across complete or repeated sentences in the appendix.

Though the paper is written in a well-organized way most of the time, notations coming out of the blue and not rigorous statements in the main algorithm make it confusing for ones who are trying to parse the algorithm and theorem to get some intuitions behind the math. Sloppy notations truly harm the clarity as well as the formality of this paper.

2. **Exponential dependency on $S$**:
It feels the exponential dependency $S^{N}$ appering in constant $C_1$ is far from tight, given the markov chain associated with each arm is independent. To compare with, the regret by [Zhou et al](https://arxiv.org/abs/2001.09390) scales linearly with $M$, which is the number of hidden states in the common MC shared by all arms. In the restless bandit setting, the complexity should be of $M$ independent MCs with S hidden states rather than one MC with $S^N$ hidden states.

**Other Concerns**
1. **Why $\sqrt{T}$? More Intuition is needed.**  I’m not fully convinced by why TSEETC is able to improve regret from $T^{2/3}$ by zhou, whose algorithm mostly resembles TSEETC, except for using the UCB estimator constructed with the spectral method for HMM. Based on comparing both algorithms and going through the proof sketch, what directly improves the bound is that TSEETC has a longer exploitation phase in each episode and thus there are only $\sqrt{T}$ episodes less than $T^{2/3}$ by [Zhou et al](https://arxiv.org/abs/2001.09390). Given both algorithms do not use the on-policy data in exploitation phase (by the way I assume it happens for TSEETC because the notation is not clear), it implies posterior sampling concentrates to the ground truth model parameter better than UCB in terms of sample efficiency. It seems kind of counterintuitive based on the understanding of TS v.s. UCB from classic bandit literature, or the bottleneck of [Zhou et al](https://arxiv.org/abs/2001.09390) is to the spectral estimator around which the UCB is constructed?

2. **Bayesian regret.** This concern relates to the previous one. A common understanding from the classic bandit literature is that UCB and TS based algorithms usually have regrets of the same order, and TS based algorithms have strictly worse regret bounds from a frequentist view. I’d like to know if it’s possible to have a UCB-based algorithm achieving \sqrt{T} regret.

3. **Computational cost of posteriors.** To compute the exact posterior, one has to exhaust all possible state transitions of length $\tau_1$, which means a total number of passes exponential to $\tau_1$,  for $\sqrt{T}$ episodes. Though $\tau_1$ would be of a constant order in theory, does this impose a higher computational cost when realizing TSEETC than SEEU in practice?



**Summary Of The Paper:**

This paper focuses on solving online restless bandits with unknown parameter and unobservable states by the proposed algorithm TSEETC. A Bayesian regret bound with $O(\sqrt{T})$ dependency is established, which matches the lower bound dependency on $T$ and improves the existing $O(T^{2/3})$ bound derived for a related problem setting by [Zhou et al](https://arxiv.org/abs/2001.09390).  Also, simulations are showing that TSEETC outperforms existing algorithms in regret as proof-of-concept.


**Summary Of The Review:**

Based on my current appreciation of the reget bound which I'm not fully convinced by and the current techinical presetation where misleading/confusing notations appear here and there, I give my recommendation as a borderline/ marginally weak rejection. I'd be more than happy to raise my score if mainly the **Weakness 2** and **Concern 1** can be addressed and cleared out, with notations being improved in the revision.

---

> ### Author Response · Authors · 2022-11-17
> **Exponential dependency on $S$ and intitution behind why $\sqrt{T}$**
>
> Thank you for your valuable comments, especially about the exponential dependency on $S$ and the intuition behind why $\sqrt{T}$ . We will address your concerns accordingly and will upload the rebuttal revision later.
>
> Q1: notations are somewhat sloppy:
>
> R1: We apologize for the notation problems. The reviewer's guesses are correct and we have corrected them in the revised version.
> For line 12, we have replaced the old notations by the more explicit ${r_{t_1: t_1+\tau_{1}},...,r_{t_k: t_k+\tau_{1}}
> 	  }$ for the reward history of all previous $k$ exploration phases, and $b_{t_1: t_1+\tau_{1}},...,b_{t_k: t_k+\tau_{1}}$ for the belief history of all previous $k$ exploration phases.
>   We have replaced $M^*$ by the true parameters $P^*,R^*$  and $M_k$ by the true parameters $P_k,R_k$.
>
> Q2: Exponential dependency on $S$
>
> R2: Thanks for this constructive comment. This $S^N$ factor is indeed very loose. In the revised version, we use the result from Proposition 4 of xiong et.al. to improve it to $SN$, as suggested by the reviewer.
>
> Q3 : Intuition about why $\sqrt{T}$
>
> R3:  As suggested by the reviewer, we think the bottleneck of Zhou et al is the spectral estimator, which has an error bound of order $1/\sqrt{k}$. To control this error, they were forced to use a shorter exploitation phase, which results in a larger regret. In our case, TS gives a better estimation bound for the average error and hence a smaller regret, but only in the Bayesian sense. It is unclear yet whether TS can improve the frequentist regret.
>
> Q4: Bayesian regret
>
> R4:  We do not yet know whether a UCB-based algorithm can achieve $\sqrt{T}$ regret. It would be interesting to investigate this in the future.
>
> Q5: Computational cost of posteriors：
>
> R5: We agree this exponential dependence on $\tau_1$ may lead to high computational cost if implemented as is. However, most of the state transitions have very small probabilities and, for practical purposes, can be ignored in the posterior update. This approximation greatly reduces the computational cost. In fact, our implementation in the experiments section uses such an approximation and achieves good performance. The theoretical analysis for the impact of such approximations is left for future work.

---

### Decision · Program_Chairs · 2023-01-20

**Decision:**

Reject

**Justification For Why Not Higher Score:**

This is a borderline paper, some assumptions are a bit strong (hence results could be better), but the writing  makes it too hard to follow. I believe that this paper will benefit a lot from polishing.

**Justification For Why Not Lower Score:**

N/A

**Metareview: Summary, Strengths And Weaknesses:**

This is an interesting paper on a specific POMDP, restless bandits with unobserved states (of arms that are not played).

The major concern that we (reviewers and myself) have are the following:
The writing, claims and notations are sometimes sloppy and very difficult to follow and this paper lacks lots of discussions. In particular, the choice of looking at Bayesian regret against the frequentist one seems quite arbitrary (and the justification is not very convincing).

The setting is new and results seem correct (yet again, the devil hides in details, and those are not always clear), so we are not 100% of them.

We hesitated, and I finally I prefer to be conservative rather than speculative which is why I, unfortunately, recommend rejection, with the following recommendation: please do take the time to polish that paper for the next submission. If you do that extra effort, I am sure this paper will be accepted (unless, of course, the devil is actually here in the details....)